# Deciphering driver regulators of cell fate decisions from single-cell transcriptomics data with CEFCON

Peizhuo Wang ⬤ [1,4], Xiao Wen ⬤ [2], Han Li ⬤ [1], Peng Lang[1], Shuya Li[1,4], Yipin Lei[1], Hantao Shu ⬤ [1], Lin Gao ⬤ [3], Dan Zhao ⬤ [1] ✉ & Jianyang Zeng ⬤ [1,4] ✉

Single-cell technologies enable the dynamic analyses of cell fate mapping. However, capturing the gene regulatory relationships and identifying the driver factors that control cell fate decisions are still challenging. We present CEFCON, a network-based framework that first uses a graph neural network with attention mechanism to infer a cell-lineage-specific gene regulatory network (GRN) from single-cell RNA-sequencing data, and then models cell fate dynamics through network control theory to identify driver regulators and the associated gene modules, revealing their critical biological processes related to cell states. Extensive benchmarking tests consistently demonstrated the superiority of CEFCON in GRN construction, driver regulator identification, and gene module identification over baseline methods. When applied to the mouse hematopoietic stem cell differentiation data, CEFCON successfully identified driver regulators for three developmental lineages, which offered useful insights into their differentiation from a network control perspective. Overall, CEFCON provides a valuable tool for studying the underlying mechanisms of cell fate decisions from single-cell RNA-seq data.

Cell fate decisions are fundamental biological processes involved in cell differentiation, reprogramming, and occurrence of diseases[1]. Recently, the advent of single-cell RNA-sequencing (scRNA-seq) has enabled the studies of cell fate decisions at single-cell resolution. By capturing the transcriptome of individual cells, scRNA-seq allows one to organize cells in pseudo-temporal order, reconstruct differentiation trajectories[1–4], and dissect transient states[5,6]. Despite these advancements, comprehending the mechanisms of how cells are controlled to determine their fates still remains a challenge.

Notably, the state of each cell is affected via an interplay between internal and external cellular signals that, in turn, perform complex transcriptional regulation events in a dynamic manner[1,7]. Therefore, finding the most critical driver factors, such as driver regulators, is crucial for understanding the control of cell fate decisions. Over the past decades, biologists have made significant progress in finding key transcription factors (TFs) associated with cell differentiation and reprogramming through experimental screening[8,9]. For instance, it has been observed that a small number of specific TFs, such as *SOX2*, *NANOG*, and *OCT4* (*POU5F1*), are sufficient to maintain the identity of embryonic stem cells and regulate their developments[10]. Recently, a number of computational approaches have been proposed to fully exploit single-cell sequencing data to facilitate the identification of important genes associated with differentiation potentials[11,12], dynamic behaviors[5,6], or specific states[13]. Nonetheless, there still remains a significant need for developing the methods that can directly elucidate the driver roles of these genes in controlling cell fates.

The gene regulatory network (GRN) is vital for controlling gene expression changes during cell lineage specification. A number of

[1]Institute for Interdisciplinary Information Sciences, Tsinghua University, 100084 Beijing, China. [2]CAS Key Laboratory of Genomic and Precision Medicine, Beijing Institute of Genomics, Chinese Academy of Sciences and China National Center for Bioinformation, 100101 Beijing, China. [3]School of Computer Science and Technology, Xidian University, 710071 Xi'an, Shaanxi Province, China. [4]Present address: School of Engineering, Westlake University, 310030 Hangzhou, Zhejiang Province, China. ✉e-mail: zhaodan2018@tsinghua.edu.cn; zengjy@westlake.edu.cn

computational methods have been developed to identify key TFs (or master regulators) that play important roles in cellular differentiation or reprogramming through interactions between TFs and their downstream target genes from bulk[14–18] or single-cell RNA-seq[13,19,20] data. However, these methods mainly suffer from two issues. First, they rely heavily on the inferred GRNs, while current GRN construction methods only based on scRNA-seq data are still not accurate enough[21]. Second, most methods focus primarily on gene expression differences between discrete states of cell types or tissue types, often leaving out the continuous dynamics of cell fate transitions[16–18], thus limiting their capacity in identifying driver regulators. For the first issue, it is essential to construct a reliable GRN related to a specific cell fate lineage. Recent powerful graph neural networks (GNNs)[22] have demonstrated promising performance in deconvoluting node relationships in graphs, which may provide an effective tool to help address this issue. For the second issue, the combination of the Waddington landscape[23] and control theory[24] can actually provide a useful framework for understanding the dynamics of biological systems. The Waddington landscape can be used to illustrate the dynamic process of a developmental system established by gene regulatory relationships. Such dynamic systems can be described using the GRN and pseudotime information inferred from single-cell sequencing data. Meanwhile, control theory can be applied to analyze how gene interactions influence the development of a biological system. Through control theory, the complex systems within cells can be modeled as GRNs, where driver nodes are defined as those critical genes that drive the entire system to a desired state through perturbations[24]. Thus, control theory may offer useful insights into the identification of driver regulators controlling cell fate decisions.

In this work, we develop CEFCON, a computational framework for deciphering cell fate control from single-cell RNA-seq data. CEFCON first employs a graph attention neural network under a contrastive learning framework to construct a cell-lineage-specific GRN. Then, CEFCON characterizes the cell fate dynamics based on the network control theory and identifies the driver regulators through combining the control-based methods with our proposed influence score, which measures gene importance in the constructed GRN. In addition, CEFCON detects the regulon-like gene modules (RGMs) involving the identified driver regulators. We

performed evaluation analyses on several benchmark datasets and demonstrated that CEFCON consistently exhibited superior performance in GRN construction, driver regulator identification, and gene module identification. Furthermore, we applied CEFCON to study the differentiation of mouse hemopoietic stem cells and reveal the driver regulators controlling three directions of cell fates. Overall, we provide a useful tool for systematically understanding the regulatory mechanisms of cell fate decisions from scRNA-seq data. CEFCON is implemented in Python as a user-friendly package and is available at https://github.com/WPZgithub/CEFCON.

## Results

### Overview of the CEFCON framework

We develop CEFCON, a network-based framework for inferring the gene regulatory relationships and characterizing their dynamics from a perspective of network control theory to identify the driver regulators of cell fate decisions (Fig. 1). Basically, CEFCON takes a prior gene interaction network and gene expression profiles from scRNA-seq data as inputs (Fig. 1a) and consists of three main components, including cell-lineage-specific GRN construction (Fig. 1b), driver regulator identification (Fig. 1c), and regulon-like gene module identification (Fig. 1d).

In our framework, we fully exploit an available global and context-free gene interaction network[25] as prior knowledge, from which we extract the cell-lineage-specific gene interactions according to the gene expression profiles derived from scRNA-seq data associated with a given developmental trajectory. More specifically, based on the prior gene interaction network, CEFCON first employs a two-layer GNN with attention mechanism to aggregate gene expression information from neighboring genes, and then captures the relationships between genes by assigning the corresponding weights to individual edges according to the obtained attention coefficients. In each GNN layer, two parallel channels are considered, i.e., an in-coming network and an out-going network, which are defined according to the directions of message-passing. The resulting feature embeddings from both in-coming and out-going networks are concatenated and then passed into the subsequent GNN layer (Fig. 2a). Such an operation is biologically meaningful for considering the importance of a gene as a regulator or target. Here, CEFCON adopts a cosine attention score[26] and takes the absolute

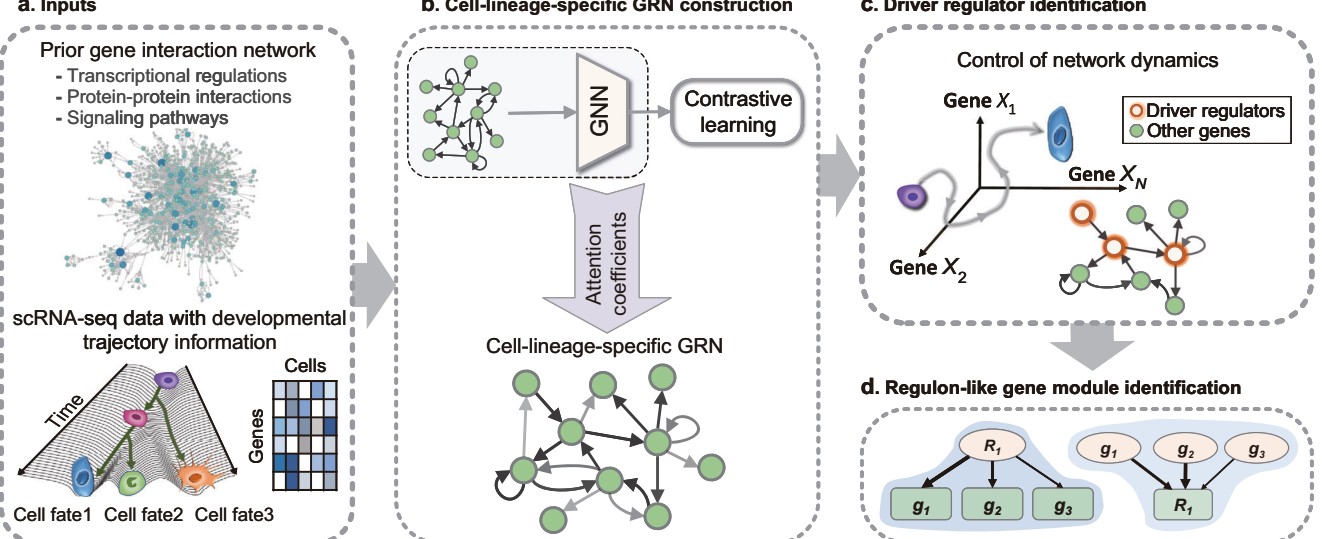

**Fig. 1 | Overview of the CEFCON framework. a** CEFCON takes a prior gene interaction network and gene expression profiles derived from the scRNA-seq data of a given cell lineage trajectory as inputs. **b** A gene regulatory network (GRN) related to a specific cell lineage is first constructed through contrastive learning on a graph neural network (GNN) with attention mechanism. CEFCON uses the attention coefficients to assign weights to individual gene interactions and then selects the top-weighted interactions to construct the cell-lineage-specific GRN. **c** CEFCON uses network control-based methods to identify driver regulators that steer cell fate decisions along the developmental trajectory. **d**, CEFCON identifies the regulon-like gene modules involving the selected driver regulators.

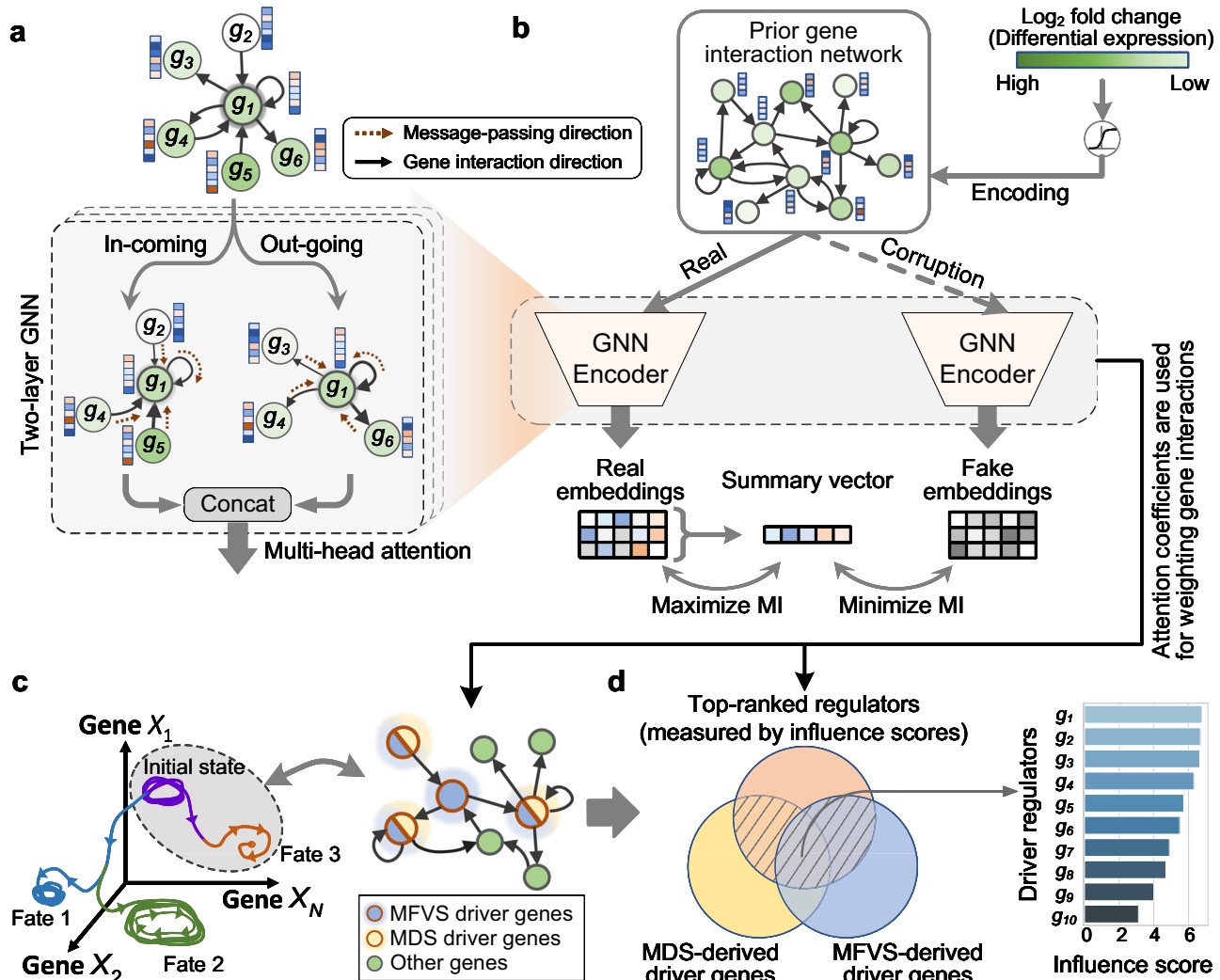

**Fig. 2 | The gene regulatory network (GRN) construction module and the driver regulator identification module. a** A two-layer graph neural network (GNN) with multi-head attention mechanism is used for GRN construction in CEFCON. In each layer, the network is divided into in-coming and out-going networks based on the directions of message-passing, which are used as two parallel channels to consider the importance of genes as regulators and targets, respectively. The outputs of both directional networks are then concatenated together. **b** A contrastive learning model used to train the GNN encoder by maximizing the mutual information (MI) between the gene embeddings and their corresponding summary. The corruption operation produces the negative samples by randomly shuffling the node features. The differential gene expression level (log2 fold change) is additionally considered

as a learnable scalar encoding. After convergence, the attention coefficients are then used to construct the cell-lineage-specific GRN. **c** The cell fate dynamics characterized by a GRN from a control theory point of view. We use two network control-based methods, i.e., minimum feedback vertex set (MFVS) and minimum dominating set (MDS), to identify the driver gene candidates on the constructed GRN. **d** The most crucial driver regulators (shaded areas in the Venn diagram) are selected among the driver gene candidates according to the influence scores for further analyses. The influence scores are defined based on the attention coefficients derived from the GRN constructer to measure the importance of each potential regulator. More details can be found in Methods.

value considering only the intensity of regulation (Supplementary Fig. 1 and Methods).

To train the GNN encoder employed in CEFCON, we adopt the deep graph infomax (DGI)[27], which relies on maximizing the mutual information (MI) between the node feature representations and corresponding graph-level representations (i.e., the summary vector of all the node feature representations), and contrastively learns the hidden feature representations by randomly shuffling the order of the input node features while keeping the network topology unchanged (i.e., the corruption operation). Based on DGI, CEFCON can learn the gene feature representations in an unsupervised manner by understanding which genes should or should not be linked. In addition, CEFCON considers the differential expression levels of genes along the developmental trajectory, guiding the attention to focus more on those genes with significant changes (Fig. 2b and Methods). Finally, the attention coefficients are scaled according to the degree of each

central node and used to construct the cell-lineage-specific GRN (Methods). Note that compared to most GRN inference methods[13,21,28–30], CEFCON constructs GRNs through focusing on transcriptional regulation and signal transduction from both TFs and non-TFs, thus making the inferred GRNs more comprehensive.

Subsequently, we use strategies based on the network control theory with nonlinear dynamics to model the cell fate dynamics at the gene level, following the assumption of the Waddington landscape[31], stating that each cell fate acts as an attractor, in which the dynamics are mainly determined by the 'roll downhill' process on a landscape and the structure is mainly dictated by the gene interactions[23]. Here, inspired by the concept of steady attractor control[32–34], which considers the naturally occurring steady states as the desirable final states, the key nodes (i.e., driver nodes) for controlling the dynamics of a network can be identified from the network structure alone (Fig. 2c). In this paper, we use two network control-based methods, i.e., minimum

feedback vertex sets (MFVS)[32] and minimum dominating sets (MDS)[34], to obtain the driver gene candidates (Fig. 2c and Methods). Since controlling a whole network usually requires a number of driver nodes[24], we further propose an influence score, which is defined according to the attention coefficients derived from the GRN constructor, to measure the importance of individual driver gene candidates (Methods). The top-ranked genes according to the influence scores among the driver gene candidates obtained from the two network control methods are regarded as the driver regulators for further analyses (Fig. 2d).

Furthermore, the regulatory dynamics is generally not driven merely by the independent actions of a few regulators but rather by regulatory sub-networks involving these regulators, such as regulons[13,18]. Here, CEFCON also extends the generally defined TF-target regulons[13,18] and detects all the gene modules of the identified driver regulators (i.e., including those non-TF genes). We refer to these gene modules as the regulon-like gene modules (RGMs), including both out-degree and in-degree types according to the regulatory roles of the corresponding driver regulators (Fig. 1d and Methods).

## CEFCON effectively constructs the cell-lineage-specific gene regulatory networks

We first assessed the quality of the cell-lineage-specific GRNs inferred by CEFCON on five benchmark single-cell RNA-seq datasets from the BEELINE framework[21], which involved seven cell lineages. These datasets included human embryonic stem cells (hESC)[35], human mature hepatocytes (hHep)[36], mouse dendritic cells (mDC)[37], mouse embryonic stem cells (mESC)[38], and three lineages of mouse hematopoietic stem cells (mHSC)[2], namely erythroid (mHSC-E), granulocyte-monocyte (mHSC-GM) and lymphoid (mHSC-L). For each lineage, we used the corresponding available ChIP-seq data as the ground-truth networks to evaluate individual constructed GRNs. For the mESC dataset, we additionally considered the loss-of-function/gain-of-function (lof/gof) data from the ESCAPE database[39] and the induced expression of TFs collected from ref. 40 for a more comprehensive assessment. For each single-cell RNA-seq dataset, we considered the top 1000 highly variable genes for evaluation.

We used the area under the precision-recall curve (AUPRC) and the early precision ratio (EPR)[21] to evaluate the performance of different GRN construction methods. Here, the AUPRC is used because it is generally more sensitive and has been considered a better metric than the receiver operator characteristic curve (AUROC)[41] when dealing with heavily unbalanced labeled data. The EPR is defined as the fraction of true positives among the top-$k$ predicted edges compared to a random predictor, where $k$ is the number of edges in the ground-truth network. We compared CEFCON with SCINET[42], NetREX[30], and CellOracle[43], three context-specific network construction methods that require prior networks, GRNBoost2[29] which uses gradient boosting to construct GRNs only based on gene expression data[21], and DeepSEM[28], a deep learning based method for GRN reconstruction from scRNA-seq data (Supplementary Note A.2). We also reported the results through randomly selecting edges from the prior gene interaction network (denoted by Random_NicheNet).

Overall, CEFCON achieved superior performances on all the benchmark datasets in terms of AUPRC (Fig. 3a), with 64% improvement compared with the second-best method (i.e., DeepSEM) and 92% improvement compared with another prior network-based method (i.e., NetREX) (Supplementary Fig. 2). In addition, CEFCON performed well in terms of EPR, achieving the best on the mDC, mESC and mHSC-E datasets and equally well with the second-best method on the other datasets (Fig. 3b). Note that the EPR values of CEFCON and NetREX were comparable to the random case (i.e., Random_NicheNet) on the ground-truth networks of both mESC and mHSC datasets. This was probably because the EPR is constrained by network densities[21] and the densities of these ground-truth networks are equal or even higher than

those of the corresponding input prior gene interaction networks (Supplementary Tables 2 and 3). Nevertheless, CEFCON still showed a clear advantage over the baseline methods in terms of AUPRC on these datasets. Moreover, we found that the methods depending on a prior interaction network (i.e., NetREX, CellOracle, Random_NicheNet, and CEFCON) performed better than those non-prior-based methods in most cases, suggesting that the input prior interaction network can provide favorable information for GRN construction. In fact, even randomly selecting edges from the prior gene interaction network (i.e., Random_NicheNet) can still obtain certain reliable results. Thus, it was reasonable to speculate that probably the high-quality and context-free prior gene interaction network was able to make up for the problems caused by the sparsity and high noise of single-cell data. Furthermore, CEFCON was robust to low levels of edge perturbation of the prior gene interaction network. For instance, when 20% edges were randomly shuffled, the AUPRC of CEFCON dropped only ~10% on average (Supplementary Fig. 3).

We found that the inferred GRNs displayed a scale-free property, i.e., their degree distributions followed a power law (Fig. 3c, d and Supplementary Fig. 4), which is the characteristic of most biological networks[44]. Notably, the average clustering coefficients of the inferred GRNs were significantly larger than those of sub-networks randomly derived from the prior gene interaction networks (Fig. 3c), which was consistent with the previous findings that the large clustering coefficients are an intrinsic feature of biological networks[44]. Moreover, we observed that individual TFs exhibited a higher number of interactions compared to non-TFs within the CEFCON inferred GRNs (Supplementary Table 4), which was consistent with the well-established knowledge in the field that TFs play critical roles in cell fate decisions and tend to have extensive regulatory interactions[8]. In brief, CEFCON can construct reliable cell-lineage-specific GRNs, which can thus provide a trustworthy basis for the downstream driver regulator identification task.

## CEFCON is able to identify the driver regulators of cell fate decisions

We next assessed the performance of CEFCON in identifying the driver regulators mainly on the mESC and hESC datasets, which contained relatively more information that can be used as ground-truth data for evaluation. More specifically, for the ground-truth data, we first used three gene sets that are associated with cell fates and ESC development from the Gene Ontology (GO) resource[45], namely the 'cell fate commitment (GO:0045165)', the 'stem cell population maintenance (GO:0019827)' and the 'endoderm development (GO:0007492)'. The last two gene sets were selected because they describe the primary features of stem cells regarding their self-renewal and pluripotency, and both mESC and hESC datasets are about the ESC differentiation into endoderm cells. In addition to these GO gene sets, we also collected a set of experimentally validated genes curated from two refs. 46,47. We used the precision of the top-$k$ predictions to quantify the accuracy, and benchmarked CEFCON against five baselines, including VIPER[18], ANANSE[17], which both predict the master regulators from bulk RNA-seq data, SCENIC[13], which discovers the key TFs and the related regulons from single-cell RNA-seq data, CellRouter[48] and CellOracle[43], which both find key TFs based on their own constructed GRNs (Supplementary Note A.2).

Our comparison results showed that CEFCON achieved superior performance on driver regulator identification over other existing methods, especially for the highest-ranked genes, which were almost perfectly identified for all the ground-truth gene sets (Fig. 4a–h). Although CEFCON did not yield the best rankings with certain $k$ values on some ground-truth gene sets, such as the mESC case for endoderm development (Fig. 4c), the gaps between its performance and the best one were quite marginal. In addition, we found that ANANSE failed to find the regulators for the 'stem cell population maintenance

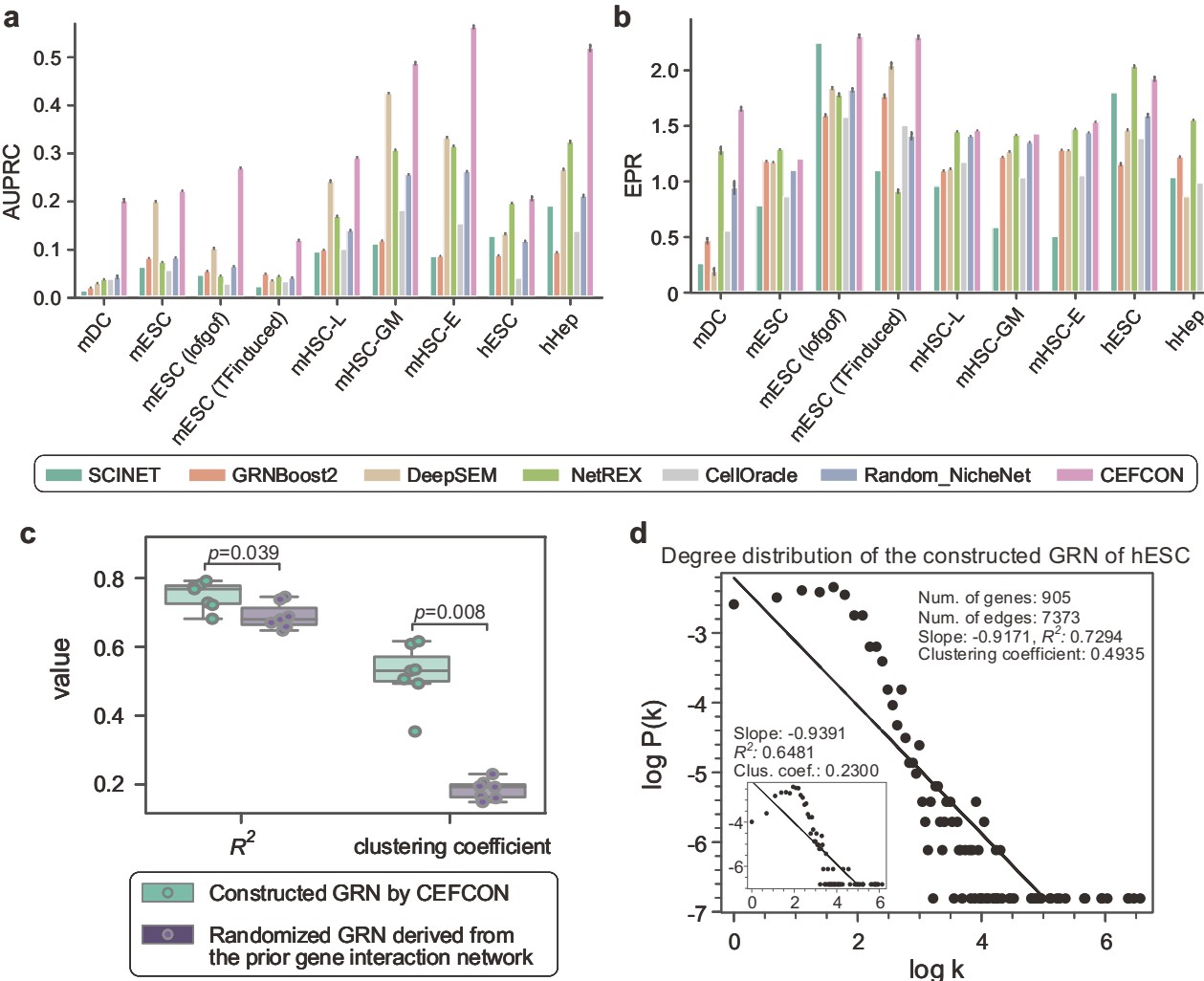

**Fig. 3 | Performance evaluation on cell-lineage-specific GRN construction.** The performance of GRN construction on different benchmark scRNA-seq datasets, measured in terms of AUPRC (**a**) and EPR (**b**), respectively. Five baselines were considered for comparison, including SCINET[42], GRNBoost2[29], DeepSEM[28], NetREX[30], CellOracle[43] and Random_NicheNet (i.e., randomly selecting edges from the prior gene interaction network). Bars and error bars signify mean ± s.d. over $n = 20$ independent experiments. **c** The comparison between the CEFCON constructed GRN and the randomized GRN derived from the prior gene interaction network, in terms of $R^2$ values and clustering coefficients. The $R^2$ is the coefficient of determination for the linear regression model to measure how close the data points are with respect to the fitted linear line. Each dot represents one of the seven datasets. Box plots show the medians (central lines) and interquartile ranges, and

the whiskers represent 1.5 × interquartile ranges. Statistical significance was calculated using the one-sided Wilcoxon signed-rank test. **d** Degree distribution and major topological properties of the GRNs constructed by CEFCON on the hESC dataset. The x-axis represents the network degree (denoted as k) and y-axis represents the frequency of the network degree k (denoted as P(k)). Both k and P(k) are log-transformed. The major network topological properties, including the number of genes, the number of edges, the slope and the $R^2$ of the degree distribution, and the average clustering coefficient, are listed in the upper right corner. The inset in the bottom left corner provides the degree distribution (also with the slope, $R^2$ and average clustering coefficient) of the randomized GRN derived from the prior gene interaction network. Source data are provided as a Source Data file.

(GO:0019827)' set and SCENIC performed poorly on the mESC dataset, although they both performed well in the other cases. On the other hand, CEFCON consistently showed satisfactory performance for all the ground-truth gene sets. In fact, the 'stem cell population maintenance (GO:0019827)' and the 'endoderm development (GO:0007492)' sets describe the start and end states of the differentiation lineage from ESCs to endoderm cells, respectively. Thus, the consistently excellent performance of CEFCON on these two gene sets implied that it can effectively discover the driver regulators along the developmental trajectory. Moreover, we found that the top-ranked genes based only on the derived influence scores had a large overlap with the driver genes obtained using the two network control-based methods (Fig. 4i), indicating that our proposed influence scores can reliably reflect the importance of driver genes in the GRNs. Furthermore, the scheme employed by CEFCON in identifying driver genes

outperformed the traditional node centrality metrics, such as degree centrality and betweenness centrality (Supplementary Fig. 5). In addition, overall CEFCON demonstrated more accurate and more robust performance in identifying driver regulators on its own constructed GRNs compared with those constructed by other methods (Supplementary Fig. 6).

We further examined the top-20 driver regulators identified by CEFCON in hESC and found that more than half of them were consistent with the ground-truth gene sets (Fig. 4j). Their gene expression trends showed significant correlations with developmental pseudotime (Supplementary Fig. 7). Besides, many genes were uniquely identified or given high rankings by CEFCON compared with those detected from other methods (Supplementary Figs. 8 and 9). In particular, three TFs of the identified driver regulators, namely *NANOG*, *SOX2*, and *POU5F1* (*OCT3/4*), which are well-known pluripotency

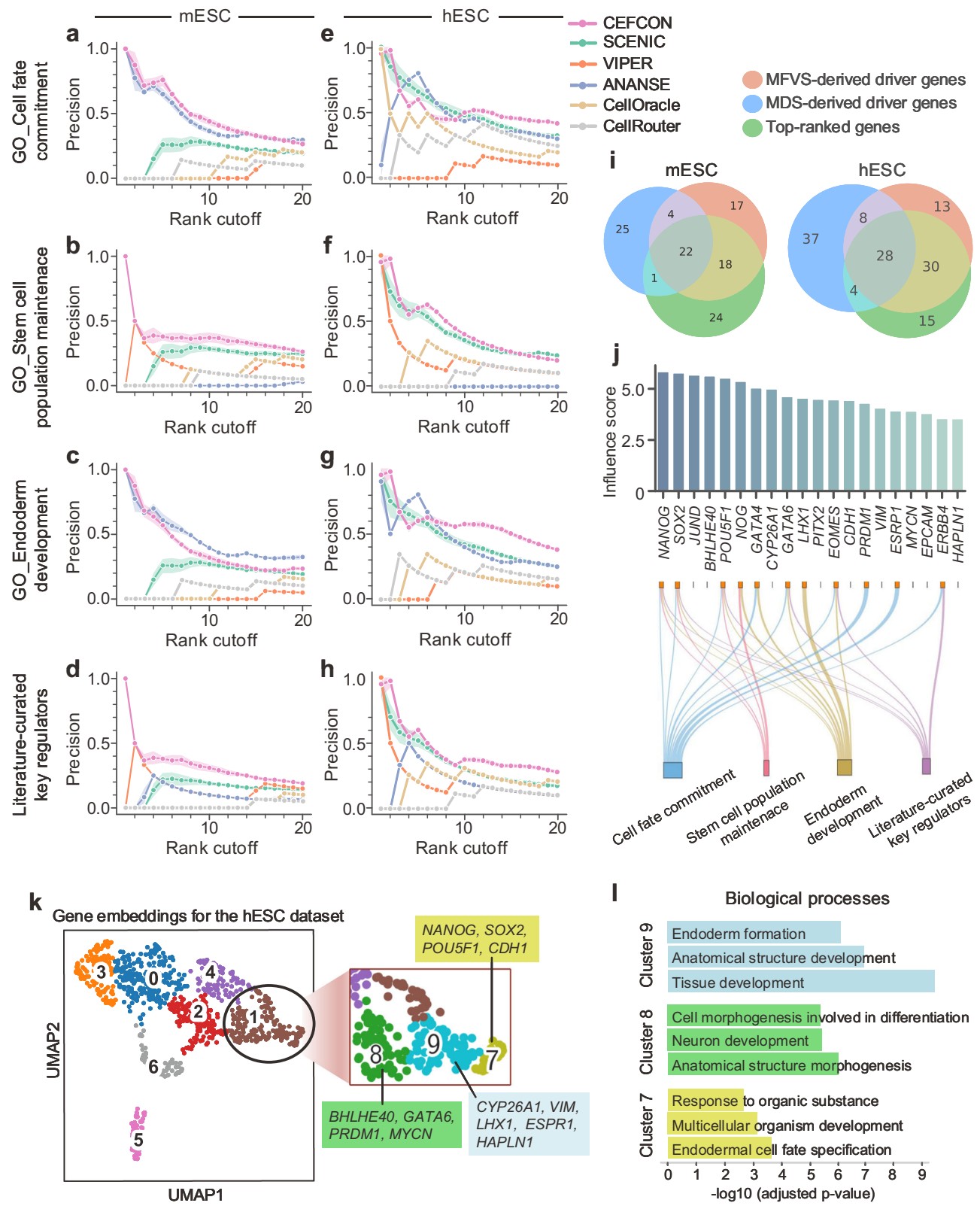

factors for human and mouse ESC development[10], were among the high rankings yielded by CEFCON (rank 1/20, 2/20, and 5/20 for *NANOG*, *SOX2*, and *POU5F1*, respectively). It had been previously reported that these TFs were mutually regulated by one another, forming cross-regulated feedforward loops[49]. In addition, another two regulators, *GATA4* and *GATA6*, which had been previously shown to induce highly selective differentiation to primitive endoderm[50], were

also identified by CEFCON with high rankings (Fig. 4j). In fact, all these TFs are pioneer factors that have been previously shown to initiate gene transcription[51] and balance the promotion or inhibition of the differentiation from ESCs to primitive endoderm cells[10,50]. Moreover, we analyzed the predicted gene interactions between *NANOG*, *GATA6*, *CDH1*, and *GATA4* based on additional ChIP-seq and epigenetic data (Supplementary Fig. 10). The results provided strong and orthogonal

**Fig. 4 | Performance evaluation on driver regulator identification. a–h** The performance on the mESC (**a–d**) and hESC (**e–h**) datasets, measured in terms of the precision of the top-*k* predicted genes among all known genes in the four ground-truth gene sets. All the results with *k* ranking from 1 to 20 were reported. The shaded area represents the variation (mean ± s.d.) of precision over 20 repeats. **i** Venn diagrams about the MDS-derived driver genes, MFVS-derived driver genes, and the top-ranked genes according to the influence scores derived by CEFCON. **j** The top-20 predicted driver regulators on the hESC dataset sorted in descending order according to their influence scores. The genes belonging to each ground-truth gene set are presented below the bar chart. **k** UMAP visualization of the gene

embeddings output by CEFCON on the hESC dataset. Gene embeddings were clustered by the Leiden method[59] with a low resolution. The top-ranked driver regulators were mainly in cluster 1, marked with a black circle. Cluster 1 is further zoomed in with a higher resolution, and the genes belonging to the top-20 driver regulators in individual sub-clusters are also marked. **l** The top enriched GO terms of the genes in sub-clusters 7 (yellow), 8 (green), and 9 (blue) in **k**, respectively. The *p*-values were measured by one-sided Fisher's exact test, adjusted for multiple hypothesis testing using the Benjamini-Hochberg false discovery rate method. Source data are provided as a Source Data file.

evidence to support the interactions predicted by CEFCON between these driver regulators.

Notably, in addition to TFs, CEFCON also identified critical non-TFs as the driver regulators (Fig. 4j and Supplementary Fig. 8). In particular, among the top-20 driver regulators identified by CEFCON, *CYP26A1*, *CDH1*, *ESRP1*, *EPCAM*, and *ERBB4* were not TFs, but they had also been reported to play key roles in ESC differentiation[52–56]. For example, the protein encoded by the *ERBB4* gene is a receptor tyrosine kinase, which had been known to regulate cell proliferation and differentiation[56]. In fact, as the initial responders of intracellular signaling pathways, receptor kinases can function as the upstream of TFs and thus play driver roles in cell fate decisions[57]. Unlike most existing methods, such as the compared methods in Fig. 4, which are limited to finding the TFs as master regulators, CEFCON can extend the identified driver regulators to a wider range of genes.

To further demonstrate the reasonableness of the driver regulators identified by CEFCON, we also used UMAP[58] to visualize the gene embeddings derived from CEFCON and clustered them using the Leiden method[59] on the hESC dataset. We found that most of the top-ranked driver regulators identified by CEFCON were distributed together (Fig. 4k). More specifically, our analysis revealed that 13 out of the top-20 driver regulators belonged to one cluster (i.e., cluster 1 in Fig. 4k), which can be further clustered into several sub-clusters, including clusters 7, 8, and 9 as shown in Fig. 4k. Notably, the three core TFs involved in the preservation of pluripotency and self-renewal of ESCs, i.e., *NANOG*, *SOX2*, and *POUSF1*, were clustered together (i.e., cluster 7). In addition, the top significant enrichment terms from the GO biological processes related to these clusters were mainly about endodermal cell fate specification, anatomical structure morphogenesis, and tissue development (Fig. 4l). All these analysis results suggested that CEFCON can provide biologically meaningful feature representations and thus imply useful information for accurately identifying the driver regulators determining cell fates.

## The regulon-like gene modules identified by CEFCON reveal relevant cell states during differentiation

We next used CEFCON to identify the regulon-like gene modules (RGMs) of the obtained driver regulators to further learn more about their regulatory roles. Here, we divided the RGMs into two types, i.e., the out-degree type and the in-degree type (Fig. 5a and Methods). In particular, we identified RGMs from the scRNA-seq datasets with annotated cell types or cell states, involving hESC, mESC, and three lineages of mHSC, as in the previous sections. To evaluate the performance in RGM identification, we benchmarked CEFCON against SCENIC[13], which constructs regulons to identify stable cell states from scRNA-seq data, from two aspects, i.e., functional enrichment analysis and cell state clustering.

We first assessed if the identified RGMs were biologically meaningful by performing their functional enrichment analyses on GO[45] gene sets and KEGG[60] pathways. We employed the precision score, which was calculated as the percentage of gene sets with at least one significantly enriched function term, to evaluate the biological importance of the RGMs (Fig. 5b, c). We also conducted a more rigorous assessment that considered the precision of gene sets with at

least 50 enriched GO terms and five enriched KEGG pathways (Supplementary Fig. 11). Overall, CEFCON substantially outperformed SCENIC on all the five tested datasets, indicating that CEFCON can capture more biologically reasonable and meaningful gene modules.

We then investigated whether the identified RGMs were able to guide the identification of cell states, as was done in the SCENIC work[13]. More specifically, we first measured the activities of RGMs on individual cells using AUCell[13] and then employed hierarchical clustering on the activity matrix of RGMs for each dataset. The normalized mutual information (NMI) was used to quantify the clustering results according to the reference labels. The comparison results showed that the RGMs derived by CEFCON were significantly better than those obtained by SCENIC (Benjamini-Hochberg adjusted p-value < 0.001) among three out of five datasets and had a comparable performance on the remaining two datasets (Fig. 5d). Moreover, as a case study, the activity heatmaps of both out-degree and in-degree types of the identified RGMs for the hESC dataset showed that the RGMs can obviously capture the relevant cell states during differentiation (Fig. 5e and Supplementary Fig. 12). In addition, we observed that although several RGMs of the top-ranked driver regulators (e.g., *CDH1*, *NOG*, *GATA4*, *PRDM1*, *SOX2*, and *JUND*) did not clearly behave as state-specific, most of them had been previously demonstrated to play persistent and vital roles during the whole differentiation process (Fig. 5e, bottom block). For instance, the transcriptional repressor *PRDM1* (also known as *BLIMP1*) has been reported to play a versatile role in controlling cell fate decisions in developing embryos and adult tissues[61]. Our results showed that *PRDM1* regulated 98 target genes (37 were repressed and 61 were activated, determined according to the negative or positive correlations of the Pearson correlation coefficient of their gene expression with that of *PRDM1*) and had high activities in both early and late states (Fig. 5f), indicating that its repressive and activate regulatory effects were highly related to the early and late states of the development, respectively. Overall, these results demonstrated that CEFCON can effectively identify the regulatory gene modules that correspond to relevant cell states during differentiation.

## CEFCON illustrates the important landmarks of cell fate decisions in mouse hemopoietic stem cell differentiation

To further demonstrate the application potential of CEFCON in deciphering cell fate decisions, we reprocessed the scRNA-seq data of mouse hemopoietic stem cells[2] and selected the 3000 most highly variable genes for in-depth analyses (Methods). As shown in Fig. 6a, b, the fates of hemopoietic stem cells (HSCs) progress through several important cell states or cell types mainly in three directions, i.e., erythroid lineage, granulocyte-monocyte lineage, and lymphoid lineage. We thus applied CEFCON to study the three lineages separately to identify the driver regulators controlling their corresponding fate trajectories.

We first selected the top-20 identified driver regulators for each cell lineage for detailed analyses (Fig. 6c and Supplementary Fig. 13). The results were reproducible across the replicates with different random seeds (Supplementary Fig. 14). We found that six genes (i.e., *Meis1*, *Gata2*, *Jun*, *Fos*, *Mycn*, and *Dusp1*), which had similar expression trends along the pseudotime, were common driver regulators in all the

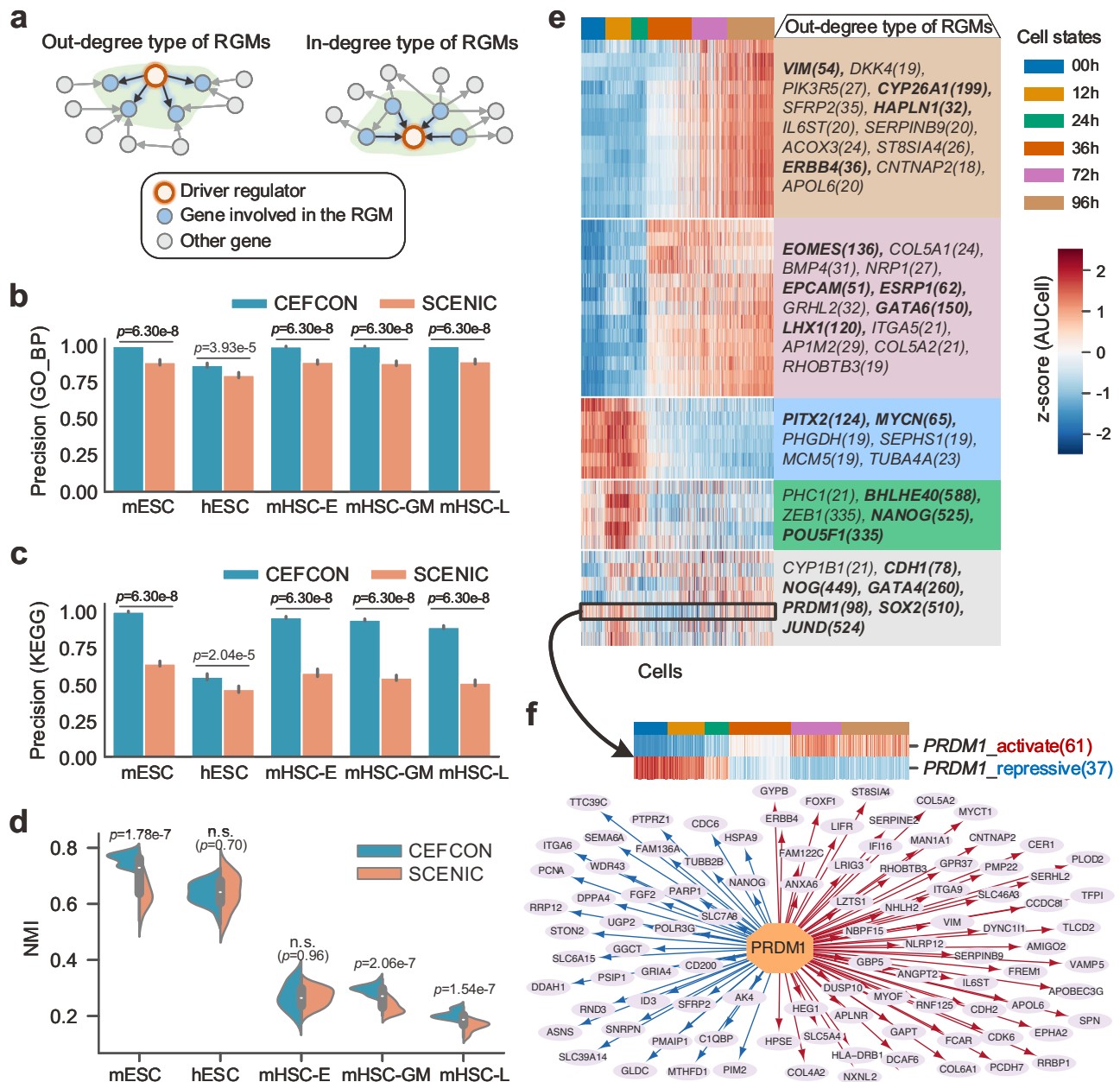

**Fig. 5 | Evaluation of the regulon-like gene modules identified by CEFCON.**
**a** Illustration of two types of RGMs identified from the in-coming and out-going networks, respectively. **b**, **c** The precision of the identified RGMs which were significantly enriched (BH-adjusted *p*-value ≤ 0.05) in GO biological process (GO_BP) terms (**b**) and KEGG pathways (**c**) on the five tested datasets, respectively. Bars and error bars signify mean ± s.d. in all panels. **d** The performance of cell clustering in terms of normalized mutual information (NMI) according to the reference labels. Each box indicates the median (central line) and interquartile range, and the whisker represents 1.5 × interquartile range. **e** The heatmap about the activities of out-degree type of RGMs in each cell for the hESC dataset. Each RGM is represented by its involving driver regulator and the number of its regulated genes is given in parentheses. Genes in bold indicate the top-20 driver regulators associated with Fig. 4j. **f** The identified out-degree type of RGM involving *PRDM1*. The repressive and activate regulatory relations are shown in blue and red, respectively (bottom subfigure), and the activity heatmaps of these two parts largely corresponded to the early state (i.e., 00h, 12h, and 24h) and late state (i.e., 72h and 96h) of the development, respectively. For **b**–**d** *n* = 20 independent computational experiments were conducted, and statistical significance was calculated using the unpaired two-sided Wilcoxon rank-sum test. n.s. not significant (*p*-value > 0.05). BH Benjamin-Hochberg. Source data are provided as a Source Data file.

three directions of cell fate commitment (Fig. 6c, d and Supplementary Fig. 15). The expression levels of these genes were high in HSCs and decreased along the differentiation process. In particular, among the six common driver regulators, *Dusp1* was the only non-TF regulator, which is a well-known proliferation-associated gene[62]. In addition, the granulocyte-monocyte lineage and lymphoid lineage shared more driver regulators (Fig. 6c), which was consistent with the classical model of hematopoietic differentiation that the lymphoid multipotent progenitors (LMPPs) may also differentiate into the granulocyte-

monocyte progenitors (GMPs)[2]. Moreover, the specific driver regulators of each lineage showed their ability to steer cell fates in particular directions, such as *Klf1* for erythroid, *Mpo* for granulocyte-monocyte, and *Mef2c* for lymphoid (Fig. 6e).

Next, we analyzed the RGM of *Gata2*, which was one of the most significant common driver regulators in the three lineages (Fig. 6f–h). We found that the identified out-degree type of RGMs had relatively large sizes in all the three lineages, in which *Gata2* regulated or interacted with over 1500 genes (Fig. 6f–h, right panels), thus implying its

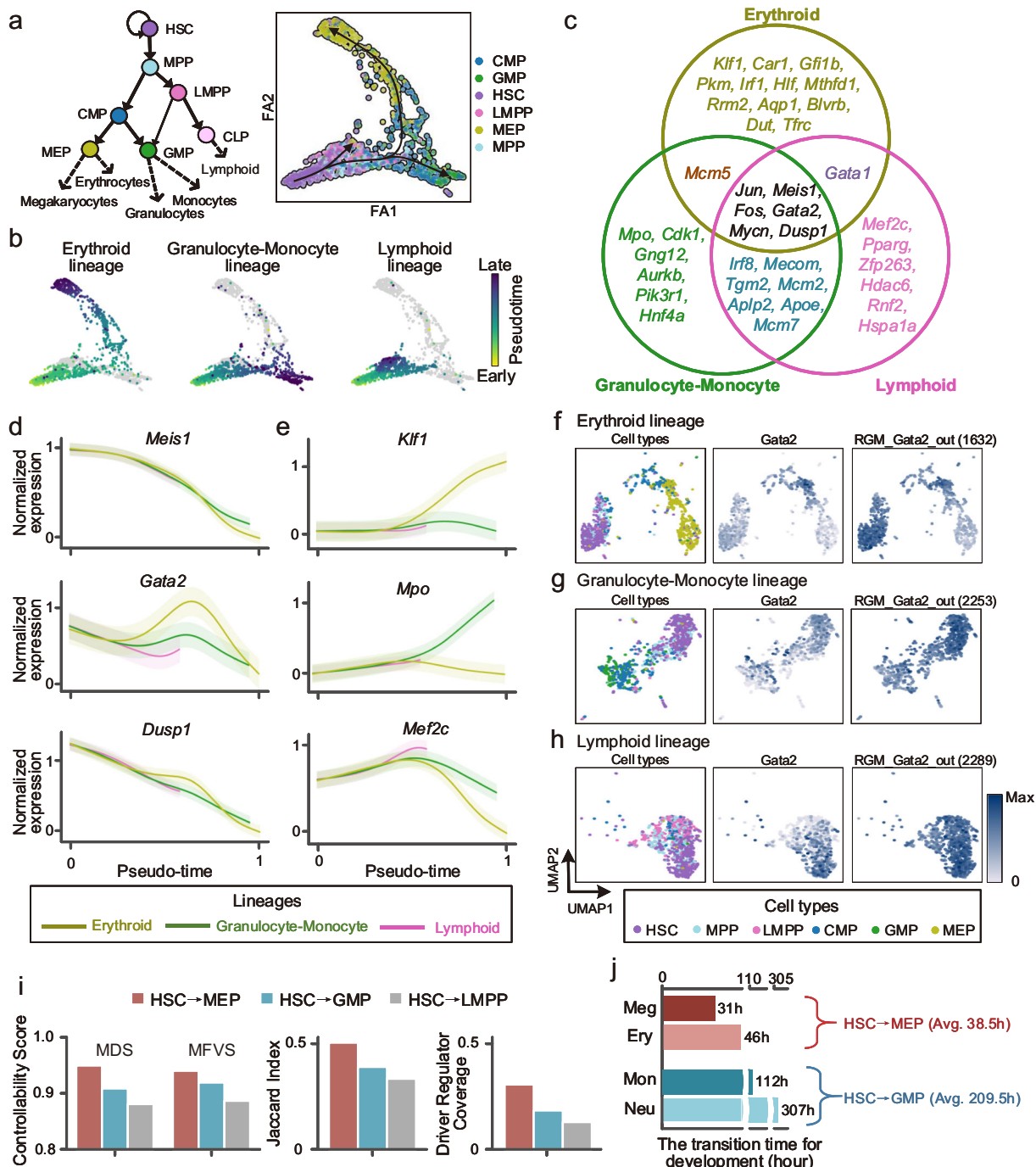

**Fig. 6 | Application of CEFCON to study the mechanisms underlying cell fate decisions in the differentiation of mouse hematopoietic stem cells. a** The classical model of the hematopoietic differentiation landscape and the force-directed (FA) visualization of three developmental trajectories derived from the scRNA-seq data. The cell type annotations were obtained from ref. 2. HSC hematopoietic stem cell, MPP multipotent progenitor, LMPP lymphoid multipotent progenitor, CMP common myeloid progenitor, MEP megakaryocyte-erythrocyte progenitor, GMP granulocyte-monocyte progenitor, CLP common lymphoid progenitor. **b** The pseudotime information and the representation of cells belonging to each cell lineage. **c** Venn diagram about the top-20 predicted driver regulators of each cell lineage. **d** Gene expression trends of three common driver regulators along pseudotime in the three cell lineages. The shaded area represents mean ± s.d.

**e** Gene expression trends of three top-ranked lineage-specific driver regulators, i.e., *Klf1* for erythroid, *Mpo* for granulocyte-monocyte, and *Mef2c* for lymphoid. The shaded area represents mean ± s.d. **f**, **h** UMAP visualization of the cell types, gene expression levels of *Gata2*, and the activities of its out-degree type of RGMs in the erythroid lineage (**f**), granulocyte-monocyte lineage (**g**), and lymphoid lineage (**h**). The number of target genes is given in parentheses behind each RGM. **i**, Network controllability assessment for the three lineages, measured in terms of controllability scores of MDS- and MFVS-derived driver genes, the Jaccard index between them, and the coverage of driver regulators. **j** The transition time of mHSC differentiation according to ref. 63. Meg megakaryocyte, Ery erythrocyte, Mon monocyte, Neu neutrophil. Source data are provided as a Source Data file.

crucial roles in regulation during HSC differentiation. Moreover, although the expression of *Gata2* fluctuated along pseudotime, its RGM activity changed gradually during the development of each lineage and was strongly specific to the HSC state (Fig. 6f–h), thus indicating its important roles in stem cell maintenance.

We further analyzed the differentiation of the three cell lineages from the perspective of network controllability. More specifically, we calculated three metrics, i.e., the controllability score of the MFVS- or MDS-derived driver genes, the Jaccard index between the driver gene sets obtained from the two control methods, and the coverage of the identified driver regulators among all the driver gene candidates (Methods). Basically, from the perspective of control theory, the larger values of these metrics generally means that it is relatively easier and thus more efficient to control the state changes of a system. For cell development, the efficiency of controlling a system can be related to the time required to reach the corresponding cell fate changes, that is, the higher values of these metrics may indicate less time required for differentiation. As shown in Fig. 6i and Supplementary Fig. 16, the percentage of driver genes controlling the differentiation from HSCs to MEPs was lower than that of the other two lineages, that is, it was more efficient to control the state changes from HSCs to MEPs, which was consistent with a previous study revealing that the time cost of differentiation from HSCs to MEPs (38.5 hours on average) was significantly less than that required from HSCs to GMPs (209.5 hours on average) under the same environmental condition[63] (Fig. 6j). In addition, according to a previous study[64], the erythroid lineage differentiates earlier than the granulocyte-monocyte lineage, whereas the lymphoid lineage differentiates later than both the erythroid and granulocyte-monocyte lineages. Such a result was also consistent with the trend of network controllability revealed in our analysis (Fig. 6i). Taken together, these results suggested that our network control-based strategy may provide new insights into the underlying mechanisms of cell fate decisions.

## Discussion

We have presented CEFCON, a network-based framework for deciphering the driver regulators of cell fate decisions from scRNA-seq data. Unlike existing algorithms for cell trajectory inference or cell fate mapping[11,12], CEFCON is a gene-level approach that fully exploits available cell lineage information to in-depth mine the key regulatory factors related to cell fate decisions.

CEFCON makes three major contributions in elucidating cell fate decisions. First, CEFCON constructs a cell-lineage-specific GRN by extracting the regulatory relationships associated with gene expression profiles from a global and context-free gene interaction network. The GRN constructor is mainly based on a graph attention neural network and a contrastive learning strategy. The graph attention neural network captures the intrinsic relationships across a global gene-gene interaction graph through propagating and aggregating the gene attributes (i.e., gene expression profiles). The contrastive learning model (i.e., DGI[27]) employed in CEFCON captures the underlying network topological structure and further enhances the interpretability of the graph attention mechanism. Second, CEFCON identifies the driver regulators that control cell fate decisions based on the constructed cell-lineage-specific GRN from a perspective of control theory. Here, the network control theory can be used for modeling the GRN dynamics that characterize cell fates, thus explaining the driver roles of the identified regulators. Third, CEFCON further detects the gene regulatory modules of the driver regulators to better reveal their roles in determining the relevant cell states during cell development.

Our benchmark tests showed that CEFCON consistently outperformed the baseline methods in all the above three aspects. In particular, CEFCON exhibited accurate and robust results in GRN construction on various scRNA-seq datasets, and accurately identified the driver regulators that control the pluripotency of stem cells and

their differentiation into endoderm cells in both human and mouse ESC datasets. In addition, CEFCON was relatively time-efficient compared to the baseline methods (Supplementary Fig. 17). Furthermore, when applied to the mouse hemopoietic stem cells, CEFCON effectively identified the driver regulators that steer cell fate changes in three directions, further elucidating their differences from a control theory perspective.

Considering the significance of discovering critical genes from GRNs, and the fact that traditional GRN construction methods from scRNA-seq data inevitably encounter the limitations of the sparse and noisy nature of single-cell data, we believe that leveraging reliable prior information can effectively help address this problem. In our study, we thoroughly explored the impact of different background networks as input prior gene interaction networks on the results. Our findings highlighted that the completeness of a prior network can affect the results of GRN construction (Supplementary Fig. 18) and driver regulator identification (Supplementary Fig. 19). Moreover, as an alternative way to the incorporation of prior information, supervised learning frameworks, such as GENELink[65], may yield superior results, but the label datasets serving as ground-truths are often lacking, especially for specific cell types or lineages.

Despite the advantages of CEFCON, there are still certain limitations. First, CEFCON can only handle one cell lineage at a time. Since cell developmental trajectories often have complex structures, such as bifurcations, it will be natural to extend the current framework to process multiple fate trajectories simultaneously. In addition, cell fates are mainly determined by the actions of various intrinsic and extrinsic cellular factors. However, our current work only focuses on intracellular factors. Future extensions can consider more extrinsic factors, such as cell–cell communication[66] and cell spatial location[67]. Furthermore, as another essential factor controlling cell fate decisions, epigenetics can also provide deep insights into understanding the complex processes associated with cell fates. By taking into account these gene-level features beyond RNA-seq data, we believe that CEFCON can also be extended to analyze single-cell data from other modalities, such as the spatial transcriptome and chromatin accessibility. Moreover, single-cell multi-omics data inherently possesses the potential to provide a more comprehensive understanding of gene regulation[68,69], representing a promising avenue for deciphering cell fate decisions.

Overall, CEFCON shows great application potential in unraveling cell fate decisions during cell differentiation and can also provide a valuable tool for analyzing other cell development processes, such as reprogramming and diseases.

## Methods

### Cell-lineage-specific GRN construction

In CEFCON, the cell-lineage-specific GRN construction process is mainly performed through a graph neural network (GNN) with attention mechanism. In addition to learning the representative embeddings of gene nodes, which is often the primary task of a GNN, here we focus more on the interpretability of the attention mechanism and make full use of it to infer the regulatory interactions for a given cell lineage.

**The graph neural network with attention mechanism.** In our GRN construction strategy, the input prior gene interaction network is represented as a directed graph $G$ with a node set $V = \{v_1, v_2, \cdots, v_N\}$, where $v_i$ represents a gene and $N$ stands for the total number of nodes. We denote the adjacency matrix of the graph $G$ as $\mathbf{A} \in \{0, 1\}^{N \times N}$ with $\mathbf{A}_{ij} = 1$ for an existing edge from $v_i$ to $v_j$ and $\mathbf{A}_{ij} = 0$ otherwise. The node features (i.e., gene expression profiles) are represented as $\mathbf{h} = [\mathbf{h}_1, \mathbf{h}_2, \cdots, \mathbf{h}_N]$, where $\mathbf{h}_i \in \mathbb{R}^F$ stands for the features of $v_i$ and $F$ stands for the dimension of node features. Given a node $v_i$, the attention-based model first learns the importance of its neighbors'

contributions to its feature representation by scoring the relationships between node features. Through applying the self-attention mechanism[70], we can calculate the attention coefficient between node $v_i$ and node $v_j$ as:

$$e_{ij} = \text{att}(\mathbf{W}_a \mathbf{h}_i, \mathbf{W}_b \mathbf{h}_j), \tag{1}$$

where att stands for an attention scoring function measuring the relevance of $v_i$ to $v_j$, $\mathbf{W}_a \in \mathbb{R}^{F' \times F}$ and $\mathbf{W}_b \in \mathbb{R}^{F' \times F}$ stand for the learnable weight matrices associated with the source and target nodes, respectively, and $F'$ stands for the dimension of latent feature representations in hidden layers. The attention coefficients between $v_i$ and its 1-hop neighbors are then normalized through a softmax function, denoted as $\alpha_{ij}$:

$$\alpha_{ij} = \text{softmax}_j(e_{ij}) = \frac{\exp(e_{ij}/\tau)}{\sum_{r \in \mathcal{N}_i \bigcup \{v_i\}} \exp(e_{ir}/\tau)}, \tag{2}$$

where $\mathcal{N}_i$ stands for a 1-hop neighborhood of node $v_i$, and $\tau$ stands for a temperature parameter. Here, the softmax normalized attention coefficients can be represented as the probabilities of considering the corresponding neighboring nodes as regulators or regulated targets. The temperature parameter $\tau$, where $\tau < 1$, in the softmax function is used to force the probability distribution to be sharper, which can help the attention focus more on a relatively small number of relevant neighbors. We followed a previous study[71] and empirically set $\tau = 0.25$ in our framework.

The output features of every node are then defined as a linear combination of its neighbors' features according to the attention coefficients. In addition, we use the multi-head attention mechanism[70] to jointly capture more views of different feature representations, followed by an activation layer. Thus, the output features of node $v_i$ at each layer can be written as:

$$\mathbf{h}'_i = \mathop{\Big\|}_{k=1}^{K} \sigma \left( \alpha_{ii}^k \mathbf{W}_a^k \mathbf{h}_i + \sum_{j \in \mathcal{N}_i} \alpha_{ij}^k \mathbf{W}_a^k \mathbf{h}_j \right), \tag{3}$$

where $\sigma$ stands for a nonlinear activation function, $K$ stands for the total number of heads, $\|$ stands for the concatenation operation, and $\alpha_{ij}$ stands for the attention coefficient as defined in equation (2).

To enhance the interpretability of the attention and make it biologically meaningful, the employed background network (i.e., the input prior gene interaction network) is further divided into an incoming network and an out-going network for considering different message-passing directions. Such an operation can help understand the importance of genes as regulators or regulated targets. Then the final output feature embeddings of each node are defined as the concatenation of the outputs from the two separate directional networks, that is:

$$\mathbf{h}'_i = \text{concat}(\mathbf{h}'_{i,\text{in}}, \mathbf{h}'_{i,\text{out}}), \tag{4}$$

where $\mathbf{h}'_{i,\text{in}}$ and $\mathbf{h}'_{i,\text{out}}$ stand for the outputs of the in-coming and outgoing networks, respectively. The complete module of GRN construction contains a two-layer GNN mentioned above. In addition, we apply batch normalization before each GNN layer and a feedforward neural network after each GNN layer. More details about the GRN construction module can be found in Supplementary Fig. 1.

**The attention scoring function.** CEFCON uses a cosine attention scoring function[26] to measure the regulatory relationship between two genes. Specifically, we use the absolute value of the cosine function to consider the strength of regulation, while ignoring whether the regulatory relationship is activated or repressed. In addition, to

emphasize the significance of those highly perturbed genes during cell development, we take into account the differential expression levels in the attention module as scaling factors. More specifically, the attention scoring function employed in CEFCON is defined as:

$$\text{att}(\mathbf{W}_a \mathbf{h}_i, \mathbf{W}_b \mathbf{h}_j) = D_j \cdot \left| \frac{(\mathbf{W}_a \mathbf{h}_i)^T (\mathbf{W}_b \mathbf{h}_j)}{|\mathbf{W}_a \mathbf{h}_i| \cdot |\mathbf{W}_b \mathbf{h}_j|} \right|, \tag{5}$$

where $|\cdot|$ stands for the Euclidean norm, $T$ stands for the matrix transpose operation, and $D_j \in (0, 1)$ stands for the encoding of the differential expression level of gene $v_j$ (see the next subsection for more details).

We also discussed the performance of GRN construction based on another two well-known attention techniques, including the scaled-dot product attention[70] and the additive attention based on a single-layer feedforward neural network[72], and provided these schemes as additional options in our package (Supplementary Note A.1 and Supplementary Fig. 20).

**Encoding the differential expression levels of genes.** To encode the differential expression levels of genes, those genes whose expressions were significantly perturbed between the start and all the following states along the developmental trajectory were identified using MAST[73]. The p-values were adjusted for multiple testing using the Benjamin-Hochberg method[74], with a false discovery rate cutoff of 0.01. Here, the developmental states were simply annotated by equally dividing cells into $K$ parts along the pseudotime trajectory. For gene $v_j$, we use the absolute values of $\log_2$ fold changes to measure its differential expression levels between the start and all the subsequent states and then take their average across all the developmental states, denoted as $l_j = (\sum_{k=2}^{K} |\log_2 \text{foldchange}|_{j,k})/(K-1)$, which is then encoded as a learnable scalar through a sigmoid function, that is:

$$\begin{aligned} D_j &= \text{sigmoid}\left( c \cdot l_j + d \right) \\ &= \frac{1}{1 + e^{(-c \cdot l_j - d)}}, \end{aligned} \tag{6}$$

where $c$ and $d$ stand for the trainable scalar parameters shared across all the GNN layers. With this encoding scheme, each gene can amplify or diminish its impact on all its neighboring genes according to the corresponding differential expression levels. We discussed the impact of incorporating differential gene expression information on GRN construction using various benchmark datasets, and the results indicated that considering this information can improve the performance of GRN construction (Supplementary Fig. 21).

**Contrastive learning based on the deep graph infomax technique.** To train the above graph attention neural network, we use the deep graph infomax (DGI) technique[27], a flexible unsupervised, and contrastive learning approach that maximizes the mutual information between the node representations and the global representation of the entire graph. More specifically, the loss function of DGI is defined as a binary cross entropy loss:

$$\mathcal{L} = \frac{1}{2N} \left( \sum_{i=1}^{N} \mathbb{E}_{(\mathbf{X},\mathbf{A})} \left[ \log \sigma \left( \mathbf{h}_i^T \mathbf{M} \mathbf{s} \right) \right] + \sum_{j=1}^{N} \mathbb{E}_{(\widetilde{\mathbf{X}},\widetilde{\mathbf{A}})} \left[ \log \left( 1 - \sigma \left( \widetilde{\mathbf{h}}_j^T \mathbf{M} \mathbf{s} \right) \right) \right] \right), \tag{7}$$

where $\mathbf{h}_i$ and $\widetilde{\mathbf{h}}_j$ denote the real and corrupted feature representations generated by the GNN encoder, respectively, $\mathbf{s} = \sigma(\frac{1}{n} \sum_{i=1}^{N} \mathbf{h}_i)$ represents the global graph-level summary, $\mathbf{M} \in \mathbb{R}^{F \times F}$ stands for a trainable scoring matrix, and $\sigma$ stands for the sigmoid function. Here, $(\mathbf{X}, \mathbf{A})$ represents the pair of real node features and network structure, while $(\widetilde{\mathbf{X}}, \widetilde{\mathbf{A}})$ stands for the pair of corrupted features and network structure

by randomly permuting node features. More specifically, we randomly assign the gene expression profile of a gene to a different gene while keeping the network topology unchanged.

**The scaled attention coefficients for weighting the regulatory relationships.** In our CEFCON framework, when the proposed graph attention neural network converges, the attention coefficients are adopted to measure the strength of relationships between genes. To make the attention coefficients comparable globally, we scale the attention coefficients $\alpha_{ij}$ by multiplying them with the in-degree or out-degree of the central node in the in-coming network or out-going network, respectively. The final interaction weight, denoted as $\beta_{ij}$, is the average of the scaled attention coefficients on both directional networks, that is:

$$
\begin{aligned}
\beta_{ij}^{in} &= \alpha_{ij}^{in} \sum_{j=1}^{N} \mathbf{A}_{ij}, \\
\beta_{ij}^{out} &= \alpha_{ij}^{out} \sum_{i=1}^{N} \mathbf{A}_{ij}, \\
\beta_{ij} &= \frac{\beta_{ij}^{in} + \beta_{ij}^{out}}{2},
\end{aligned}
\tag{8}
$$

where $\mathbf{A}$ stands for the adjacency matrix of the input graph $G$, $\beta_{ij}^{in}$ and $\beta_{ij}^{out}$ stand for the scaled attention coefficients obtained from the in-coming and out-going networks, respectively.

We use the scaled attention coefficients as weights to rank the edges of the prior gene interaction network, and then select the top-ranked edges to construct the cell-lineage-specific GRN. Since a two-layer GNN is used in this work, we combine the scaled attention coefficients of the first layer (i.e., $\beta_{ij}^{(1)}$) and the second layer (i.e., $\beta_{ij}^{(2)}$) via $\mu\beta_{ij}^{(1)} + (1-\mu)\beta_{ij}^{(2)}$, where $\mu \in [0,1]$ stands for a tunable parameter.

**Edge selection for cell-lineage-specific GRN construction.** We choose at most $k_d N$ edges according to their final scaled attention coefficients to derive the cell-lineage-specific GRN, where $k_d$ stands for the average degree of the constructed GRN and $N$ stands for the total number of genes. We analyzed the effect of the parameter $k_d$ on the performance of driver regulator identification, and then set $k_d = 8$ for all the computational experiments to ensure good performances while maintaining reasonable network sizes (Supplementary Figs. 22 and 23).

**Statistics and reproducibility.** Our GRN construction takes the log-transformed and scaled gene expression profiles as the input gene features. For each scRNA-seq dataset, to take into account the dataset-specific gene relationships[75], we first supplemented the prior gene interaction network by adding the top 1% of gene co-expression associations (Spearman's correlation coefficients > 0.6). The encoder of GRN construction consists of a two-layer graph attention neural network with four heads and hidden sizes of 128 for all the GNN layers, GELU nonlinear activation[76], batch normalization before each GNN layer, and a feedforward neural network after each GNN layer (Supplementary Fig. 1). The size of output feature embeddings is set to 64. We optimized the model using an Adam optimizer with a learning rate of 1e-4 and a weight decay of 5e-4. The parameter $\mu$ for balancing the attention coefficients between the first and second layers of GNN is set to 0.5. Our further analysis showed that this parameter was relatively stable for different settings (Supplementary Fig. 20). The number of training epochs is set to 350 by default and all the results were averaged over 20 repeats to avoid the effect of randomness.

**Driver regulator identification based on network control with nonlinear dynamics**
The dynamics of activity $x_i(t)$ for gene $v_i$ evolving over time in a GRN can be written as a system of ordinary differential equations (ODEs),

that is:

$$
\dot{x}_i = F_i(\mathbf{x}), i = 1, 2, \cdots, N,
\tag{9}
$$

where $F_i$ stands for a nonlinear function depending on the regulatory relationships[33] and $N$ stands for the total number of genes in the GRN. Mochizuki et al.[33,77] have proved that if the nonlinear function $F_i$ satisfies only a few conditions (e.g., continuous differentiability, dissipative and decaying), the dynamics of the system can be solely dependent on the network topology. In fact, these conditions can be easily met in most biological systems with naturally occurring end states, which correspond to cell fates[32,33,77]. Using only the network structure, our goal is to find a minimum set of driver variables (i.e., driver genes) that can fully control the dynamics of a system represented by the GRN. In this paper, we employ two classical network control-based methods, i.e., the minimum feedback vertex set (MFVS) and the minimum dominating set (MDS), to find the driver genes. Below we will describe more details about these two approaches.

**The minimum feedback vertex set method for driver gene identification.** The first network control-based method for driver gene identification is based on the feedback vertex set (FVS)[33,77] with non-linearities:

$$
\begin{aligned}
\dot{x}_i = F_i(\mathbf{x}) &= F_i\left(x_i, \mathbf{x}_{I_i}\right), i = 1, 2, \cdots, N, \\
s.t. \ &\frac{\partial F_i\left(x_i, \mathbf{x}_{I_i}\right)}{\partial x_i} < 0,
\end{aligned}
\tag{10}
$$

where $I_i$ stands for the set of predecessor nodes of gene node $v_i$ (i.e., the genes that regulate the gene $v_i$) and the constraint is a decay condition.

According to the FVS-based method proposed by Mochizuki et al.[33] and Fiedler et al.[77], controlling all the nodes in the FVS is sufficient to drive the system to any of its attractors (i.e., cell states). Here, we used the extended FVS-based method proposed by Zañudo et al. [32], which control all the source nodes (i.e., the nodes with in degree 0) and the nodes in the FVS. In graph theory, the FVS problem aims to find a subset of nodes in the graph such that the removal of these nodes leaves the graph without feedback loops. The FVS is suitable for modeling GRNs since natural biological circuits frequently contain many positive or negative feedback loops. Here, we aim to find the minimum feedback vertex set (MFVS), which can be formalized as the following 0–1 integer linear programming (ILP) optimization problem:

$$
\begin{aligned}
\min \ &\sum_{i=1}^{N} y_i, \\
s.t. \ &z_i - z_j + N y_i \geq 1, \forall \mathbf{A}_{ij} = 1, \\
&z_i \in \{1, 2, \cdots, N\}, \forall v_i \in V, \\
&\text{and } y_i \in \{0, 1\},
\end{aligned}
\tag{11}
$$

where $\mathbf{A}$ stands for the adjacency matrix of the corresponding graph (i.e., the constructed GRN), $z_i$ and $z_j$ are auxiliary variables for $v_i$ and $v_j$, respectively, and the solution $y_i = 1$ if the node $v_i$ belongs to the optimal feedback vertex set and $y_i = 0$ otherwise.

Equation (11) is an NP-hard problem that is unlikely to be solved efficiently by a polynomial algorithm. In this work, we first applied a graph contraction strategy proposed by Lin and Jou[78] and then solved the resulting ILP problem on the simplified graph using the Gurobi optimizer (https://www.gurobi.com/).

**The minimum dominating set method for driver gene identification.** The second network control-based method for driver gene identification is based on the minimum dominating set (MDS) approach[34], which aims to construct the smallest subset of nodes in a graph, i.e., the dominating set, such that every node in the graph either belongs to

the dominating set or is a neighbor of any member in the dominating set. Although the MDS-based control method is initially used for undirected networks, it can be easily adapted for a directed network by assuming that each node in the MDS only controls all of its successor neighbors. The MDS problem for directed networks can be formalized as the following 0–1 ILP model:

$$\min \sum_{i=1}^{N} y_i$$
$$s.t.\, y_i + \sum_{j \in \mathcal{N}_i} y_j \geq 1,\, \forall v_i \in V, y_i \in \{0,1\}, \quad (12)$$

where $\mathcal{N}_i$ stands for the predecessor neighbors of node $v_i$, the solution $y_i = 1$ if $v_i$ belongs to the optimal dominating set and $y_i = 0$ otherwise.

The above MDS problem is also an NP-hard one. Similar to the strategy used for solving the MFVS problem, we first used an efficient graph reduction method proposed by Ishitsuka et al.[79] for simplifying the graph and then solved the resulting ILP problem using the Gurobi optimizer.

The above two methods obtain the driver genes from different views, which are both related to topologically important nodes in the network. In this work, we use the union of driver genes obtained by these two methods as a candidate set of driver regulators.

**The gene influence scores for ranking the driver gene candidates.** As controlling an entire network usually requires a number of driver nodes[24], we also define an influence score to directly measure the importance of individual driver genes from a constructed cell-lineage-specific GRN. Since the derived attention coefficients can well reflect the likelihood of a gene being selected as a regulator in the in-coming network or as a target in the out-going network, we define the influence score of a gene as the log-transformed sum of the scaled attention coefficients of the gene's neighbors. More specifically, we first calculate the influence scores $S_i^{in}$ and $S_i^{out}$ for gene $v_i$ on the out-going and in-coming networks, respectively, and then calculate the final influence score (i.e., $S_i$) through their linear combination as follows:

$$S_i^{in} = \ln(1 + \sum_{j \in \mathcal{N}_i^p} \beta_{ji}^{out}),$$
$$S_i^{out} = \ln(1 + \sum_{j \in \mathcal{N}_i^s} \beta_{ij}^{in}), \quad (13)$$
$$S_i = \lambda S_i^{out} + (1 - \lambda) S_i^{in},$$

where $\mathcal{N}_i^p$ and $\mathcal{N}_i^s$ stand for the sets of predecessor and successor neighbors of node $v_i$, respectively, and $\lambda \in [0, 1]$ stands for a parameter balancing these two terms. We also analyzed the impact of this parameter (Supplementary Fig. 24), and empirically set $\lambda = 0.8$ to consider more significant influence from $S_i^{out}$ because the important regulators (e.g., TFs) typically have large out-degrees in a GRN[16,17]. In our CEFCON framework, the top 100 genes ranked according to the influence scores are first selected, and their overlap with the driver gene candidates obtained from the two network control-based methods is chosen as the final list of driver regulators for further analyses.

**Regulon-like gene module identification**
As we divide the input prior gene interaction network into in-coming and out-going networks based on the directions of message-passing in the employed GNN, two kinds of gene modules can be identified: (i) a gene module consisting of a regulator and its target genes, referred to as an out-degree type of RGM (e.g., TF-target regulon[13,18]), and (ii) a gene module consisting of a target and other genes that co-regulate it, referred to as an in-degree type of RGM. In this work, these two types of RGMs associated with the identified driver regulators were detected. In addition, we only kept the significantly differentially expressed genes and removed those RGMs with sizes less than ten. In the end, the

activity of an individual RGM in each cell was measured by AUCell[13] using the pySCENIC package[80].

**Assessment metrics for network controllability analyses**
The following metrics, including controllability score, Jaccard index, and driver regulator coverage, are used to measure the difficulty level of controlling a network:

$$\text{Controllability Score} = 1 - \frac{\|D_x\|}{\|V\|}, \quad (14)$$

$$\text{Jaccard Index} = \frac{\|D_{MFVS} \bigcap D_{MDS}\|}{\|D_{MFVS} \bigcup D_{MDS}\|}, \quad (15)$$

$$\text{Driver Regulator Coverage} = \frac{\|R\|}{\|D_{MFVS} \bigcup D_{MDS}\|}, \quad (16)$$

where $D_x \in \{D_{MFVS}, D_{MDS}\}$, $D_{MFVS}$ and $D_{MDS}$ stand for the driver gene sets identified using the MFVS and MDS methods, respectively, $V$ stands for the set of all genes in the network, and $R$ stands for the final list of driver regulators identified by CEFCON. Basically, the higher these metrics are, the greater the likelihood of cell fate change. More specifically, a higher controllability score means that a fewer number of driver genes are required to control the network, a higher Jaccard index represents higher consistency between the driver gene sets obtained from different methods, and a higher driver regulator coverage indicates that the list of the identified regulators covers more driver gene candidates.

**The prior gene interaction network**
We adopted a highly comprehensive gene interaction network proposed in NicheNet[25] as our prior network. The original network from NicheNet provides a collection of ligand-receptor, intracellular signaling, and gene regulatory interactions from over 50 public data sources of mouse and human. In this paper, the ligand-receptor interactions between cells were removed because we only focused on the gene interactions within individual cells. We directly used the unweighted version of the integrated network and processed it to be directed by simply treating the undirected edges as bidirectional edges. The original NicheNet gene interaction network was given in human gene symbols. To obtain a gene interaction network for mouse, we mapped the gene symbols using the one-to-one orthologs from ENSEMBL[81] and excluded those ambiguous genes. Finally, we obtained a prior gene interaction network with 25,332 genes and 5,290,993 edges for human, and 18,579 genes and 5,029,532 edges for mouse. Additionally, we selected four global and context-free gene interaction networks, including Harmonizome[82], InWeb_InBioMap[83], PathwayCommons[84], and Omnipath[85], as the alternatives for the input prior gene interaction networks (Supplementary Note A.3 and Supplementary Table 1).

**scRNA-seq data preprocessing and lineage inference**
For the in-depth analyses of mouse hematopoietic stem cell differentiation, the scRNA-seq data were preprocessed using the SCANPY package[86]. The cells with more than 200 zero expressed genes were deleted and the genes expressed on fewer than 5 cells were removed. The top 3000 highly variable genes were selected using the 'cell_ranger' method in the SCANPY[86]. Finally, the expression values were log-normalized. The lineages and pseudotime information were calculated using Slingshot[87]. The plots of the gene expression trends along the pseudotime were obtained using Palantir[11], which fitted the data into a general additive (GAM) model for each lineage.

## The GO and KEGG pathway enrichment analyses

The GO and KEGG pathway enrichment analyses were performed using gProfiler[88]. The GO terms were restricted to the categories of "biological processes". Only the significantly enriched terms that covered at least 20% genes in the gene set were considered for evaluation and further analysis. The enrichment results with false discovery rates less than 0.05 based on the Benjamin-Hochberg test[74] were considered significant.

## Reporting summary

Further information on research design is available in the Nature Portfolio Reporting Summary linked to this article.

## Data availability

All the datasets analyzed in this study are publicly available. The prior gene interaction network was from NicheNet[25], which can be downloaded from https://github.com/saeyslab/nichenetr. The scRNA-seq datasets are available in the Gene Expression Omnibus (GEO) under accession codes: GSE75748 [https://www.ncbi.nlm.nih.gov/geo/query/acc.cgi?acc=GSE75748] (hESC), GSE81252 [https://www.ncbi.nlm.nih.gov/geo/query/acc.cgi?acc=GSE81252] (hHep), GSE98664 [https://www.ncbi.nlm.nih.gov/geo/query/acc.cgi?acc=GSE98664] (mESC), GSE48968 [https://www.ncbi.nlm.nih.gov/geo/query/acc.cgi?acc=GSE48968] (mDC) and GSE81682 [https://www.ncbi.nlm.nih.gov/geo/query/acc.cgi?acc=GSE81682] (mHSC). The ChIP-seq and loss-of-function/gain-of-function (lofgof) data for validating the constructed GRN were obtained from BEELINE[21]. The gene expression response data after the forced induction of TFs in mESCs for validating the constructed GRN are available in the GEO database under accession code GSE31381 [https://www.ncbi.nlm.nih.gov/geo/query/acc.cgi?acc=GSE31381]. The TF lists of human and mouse for GRN evaluation and analyses are available at the GitHub of pySCENIC (https://github.com/aertslab/pySCENIC/tree/master/resources). The three GO gene sets, including GO:0045165 [https://www.informatics.jax.org/go/term/GO:0045165], GO:0019827 [https://www.informatics.jax.org/go/term/GO:0019827] and GO:0007492 [https://www.informatics.jax.org/go/term/GO:0007492], for evaluating the identified driver regulators are available at http://www.informatics.jax.org/vocab/gene_ontology. The lists of the literature-curated key regulators about ESC were directly obtained from refs. 46, 47 for both human and mouse. All the data used in this study are available at Zenodo (https://doi.org/10.5281/zenodo.7564872). Source data are provided with this paper.

## Code availability

The CEFCON algorithm is implemented in Python. The source code of CEFCON is available at https://github.com/WPZgithub/CEFCON[89].

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

## Acknowledgements

This work was supported in part by the National Natural Science Foundation of China(T2125007 to JZ, 32270640 to DZ), the National Key Research and Development Program of China (2021YFF1201300 to JZ), the New Cornerstone Science Foundation through the XPLORR PRIZE (JZ), the Turing AI Institute of Nanjing, the Research Center for Industries of the Future (RCIF) at Westlake University (JZ) and the Westlake Education Foundation (JZ). We thank Dr. Yuxuan Hu and Dr. Chenxing Zhang for their insightful discussions. We thank Dr. Dacheng Ma for his constructive comments on the revised manuscript.

## Author contributions

P.W. and J.Z. designed and developed the method. D.Z. and J.Z. supervised and conceived the project. P.W. designed and implemented the CEFCON algorithms, analyzed the data, and performed the performance evaluation task. X.W. helped with the data analyses and discussed the results. P.L., X.W., H.S., and Y.L. contributed to the method benchmarks. H.L., S.L., L.G., and D.Z. contributed to the experimental analyses. P.W. and J.Z. wrote the manuscript with help from other authors. All authors have reviewed and approved the final manuscript.

## Competing interests

J.Z. is a founder of Silexon and has an equity interest. All other authors declare no competing interests.
