## [Peer Review File · Nature Communications]

Deciphering driver regulators of cell fate decisions from single-cell transcriptomics data with CEFCONReviewer #1 (Remarks to the Author):

In the manuscript entitled "A network-based framework for deciphering driver regulators of cell fate decisions from single-cell RNA-seq data", Peizhuo et al. developed CEFCON, which employed a graph attention neural network under a contrastive learning framework to construct a cell-lineage-specific gene regulatory network (GRN). Firstly, the use of graph neural network with attention mechanism is creative and suitable, which is meaningful for considering the importance of genes. Secondly, the authors applied CEFCON to the data during the differentiation of mouse hematopoiesis, which could elucidate the lineage differences from the control theory. Thirdly, CEFCON could comprehensively construct GRNs from both TFs and non-TFs. Overall, CEFCON is a valuable framework to build GRN and find key genes, but some minor comments are still needed and listed below.

1. CEFCON adopted a gene interaction network proposed in NicheNet as a prior network, which models intercellular communication by linking ligands to target genes. It will be more convincing if the authors can test GRN inferred from other tools.
2. What is the computational efficiency of CEFCON?
3. There are some methods that were not included in the comparison, such as CellOracle (Nature, 2023) and some methods shown in the BEELINE framework. The authors are supposed to include the evaluation of these methods or give reasons for exclusion.
4. CEFCON performed best in terms of AUPRC (Fig. 3a), but the real values of AUPRC in all methods are low. And the benchmark results in BEELINE showed slightly higher scores of AUPRC. The authors should explain this or choose more suitable evaluation methods.
5. The authors used three gene sets that are associated with cell fates as ground truth (Fig. 4a-4h), which seemed a little bit subjective, more evaluation could be included. And as said in the introduction, "it is generally challenging to distinguish whether these genes are marker or driver genes for controlling cell fates", the authors should further explain why CEFCON identified are "driver genes" rather than "marker genes".
6. It will be more convincing if the authors can analyze novel interactions or driver genes that CEFCON predict while other framework failed.
7. CEFCON constructs GRNs by focusing on transcriptional regulation and signal transduction from both TFs and non-TFs, which is very comprehensive. I was wondering if TFs have more interaction than non-TFs in the network CEFCON inferred. And the author could give examples of several regulation patterns that do not require TF.

Reviewer #2 (Remarks to the Author):

Wang et al. described an algorithm to infer gene regulatory networks (GRNs) by integrating prior protein interaction data, single-cell RNA-sequencing (scRNA-seq) and a deep learning framework based on graph convolutional networks with attention mechanisms. The algorithm, CEFCON, predicts cell or state-specific GRNs based on contrastive learning, and applies control theory to rank critical regulatory nodes, as well as their regulons based on the inferred GRNs. This approach is flexible with regards to the identification of regulators that are transcription factors (TFs) and non-TFs. The resulting regulons capture expected biological processes. The authors applied CEFCON to scRNA-seq datasets profiling embryonic stem cells and hematopoietic cell types, and found regulators and biological processes previously reported in the literature. Overall, the methodology is interesting and by comparing predicted cell or state-specific GRNs to ChIPSeq gold standards of the respective cell lineages, the authors show that CEFCON outperforms competing methods in terms of AUPRC. However, I have some concerns:

- 1 - It is not clear how CEFCON translates into novel biological insights. The authors seem to only focus on known regulators or biological processes, which provides some confidence that the method works, but do not explain any potential new insights or gene programs/regulons that could be uniquely identified by CEFCON and NOT identified by other methods, which then would translate to new biological insights, specially in the presence of experimental follow-up.
- 2 - The high computational demand needed to run CEFCON, which according to their github page,

ideally requires processing in graphics processing units (GPU), warrants the authors to better describe how their method could identify new regulators and their regulons (therefore, new biology) when compared to other methods. If CEFCON does contribute to identifying novel biology relative to other methods (which typically demand lower computational resources). In the absence of these comparisons, although the approach taken by the authors is interesting, it does not seem that CEFCON represents a substantial advance over prior methods, especially because there are no experimental follow-ups of CEFCON predictions and their implications for cell differentiation.

3 - In the introduction, lines 19-22, the authors indicate that algorithms to infer pseudo-temporal order of cells from scRNA-seq neglect how intracellular factors control cell fate changes. I believe this is not totally true as currently available algorithms directly or indirectly identify such intracellular factors controlling cellular differentiation. Once a developmental order is identified, virtually all papers aim to identify gene programs controlling these transitions, although many do not necessarily include a GRN inference step. For instance, Lummertz da Rocha et al, Nature Communications 2018, explicitly reconstructs gene regulatory networks directly from scRNA-seq data, and establishes a score that integrates the GRN with the kinetic profiles of genes during the inferred trajectory, prioritizing transcriptional regulators (not only transcription factors) that modulate cell fate decisions. Another example is the work by Alsinet et al, Nature Communications, 2022, which uses scVelo to identify transcriptional regulators involved in myelopoiesis. There are many other examples and strategies to identify genes that control cell fate decisions. Therefore, this seems an overstatement and authors should focus on what their method is designed to do (GRN inference), instead of what is a downstream application (trajectory inference) of what they aim to achieve with their work.

4 - Although CEFCON is based on prior protein and regulatory interaction data to infer GRNs, which likely reduces false-positive interactions, it also implies that new regulatory interactions might not be uncovered, when compared to that possibility when using de novo GRN inference methods (which, of course, have a higher false-positive rate). It is not surprising that CEFCON performs better in terms of AUPRC since it is based on prior gene/protein interaction network data. The fact that CEFCON needs to be applied separately for each cell type or state present in the dataset makes the analysis workflow cumbersome.

Minor comments:

5 - It is not clear how node centrality was calculated

6 - It is not clear what are the conceptual differences between CEFCON, DeepSem, and competing methods.

7 - It is not clear how directionality of message-passing (Line 90) gene-gene interactions was defined to create out-degree and in-degree types of networks used during training and subsequently to identify regulatory interactions?. Was this information available in the prior interaction data?

8 - The author's tutorial in github should be improved to provide walk users through the analysis workflow, including output figures, how to interpret them, and how to possibly generalize the workflow for multiple cell types or states. The command line-only based approach is often not enough for users to confidently use the tool.

9 - Line 98: corresponding high-level summaries, WHICH ARE? Need to define what are input node features?

10 - Line 204: "the gaps between its performance and the best one were quite marginal." This is true overall for cases in which CEFCON provides slightly better predictions, or when other methods produce slightly better predictions. Therefore, unless the authors systematically determine what is uniquely identified by CEFCON and not by the other methods, the manuscript as it is does not represent a substantial advance over prior work.

11 - Line 286-289: This is an interesting observation. What is Figure 5f heatmap showing though? How were the active and repressive genes determined? Do any of the genes identified as repressive or active in this section actually show this behavior experimentally in other papers? To be more specific: the genes themselves identified by CEFCON (and not by other methods) were experimentally tested to have such repressive or activation roles during differentiation?

12 - Line 301: It would be interesting to see the CEFCON-repredicted HSC regulators.

13 - The role of Gata2 and their kinetics during differentiation is overall well described during HSC differentiation. It feels like authors put too much emphasis on CEFCON-predictions that are already known, instead of systematically define what is unique to CEFCON.

Overall, the authors developed an interesting algorithm based on graph neural networks to infer gene regulatory networks from single-cell RNA-sequencing data. However, it is not very clear from the paper what is the main conceptual advances of CEFCON, what new biology CEFCON can uncover despite having higher performance in terms of AUPRC, especially because improvements in performance relative to known gene sets were only marginally improved relative to other methods.

Reviewer #3 (Remarks to the Author):

This paper presents CEFCON, a graph attention based method for inferring a lineage-specific gene regulatory network (GRN) from single cell RNA-seq data. The inferred network is analyzed using node prioritization methods based on network control theory to identify potential cell-fate drivers and is also analyzed to identify gene modules centered around the predicted drivers. CEFCON works by taking as input a non-specific graph, which is further augmented with co-expression edges and the cell line-specific scRNA-seq data as node/gene attributes and uses Deep Graph Infomax (DGI) to learn gene-level embeddings and attention coefficients. The attention coefficients are then used to infer edge weights which is used to either produce a ranking of edges in a lineage-specific manner or select the top $k_d * N$ edges to obtain a network. CEFCON is evaluated on datasets available in the BEELINE benchmark dataset and was shown to have favorable performance compared to existing methods. The authors further demonstrate the driver regulator identification and gene module identification in the hematopoietic differentiation lineage. Overall, this is a clearly written paper with the main novelty being the use of graph attention to infer a context-specific network. However, the ability of CEFCON to truly model cell lineage dynamics seems limited and as such seems to be an approach that filters out edges that might be relevant in a particular condition. Accordingly the compared methods might not be most appropriate. There are also a few issues in terms of the input network, the architecture of the model and also evaluation of cell fate driver predictions that need further clarification. Though the GAT-based approach is interesting, the impact of this approach to infer cell lineage-specific gene regulatory networks seems limited. More detailed comments follow:

1. The overall concept of CEFCON is to remove edges that are not supported by the input scRNA-seq dataset to infer what is referred to a lineage-specific network. This makes the final output depend upon the completeness of the input network, though they augment it with co-expression networks inferred from the same input scRNA-seq dataset. It would be important to show how much the expression versus the input network is playing a role and how much the completeness of the input network influences the results.

2. The idea of selecting out edges has been used using linear regression settings from single cell RNA-seq (e.g. in CellOracle and Dictys) as well as for bulk data. The authors need to compare to these methods to fully demonstrate the value of their approach. The methods compared either do not use an input network or do it in a way that is much more flexible (e.g. NetRex, which uses the network as a prior), but has the issue of adding potentially more noisy edges.

3. It seems the authors used NicheNet's network as the input network which was constructed

primarily for inferring cell-cell communication networks. Since the authors are interested to infer gene regulatory networks, it might be better to focus on input networks that are directed, e.g. signaling and regulatory, and not include protein-protein interactions.

4. It is also unclear if the authors are utilizing the ligand-target networks or the prior network that was inferred by collecting interactions from KEGG, Ramilowski et al., Omnipath, PathwayCommons, InWeb_InBioMap, and other models. If the full prior network was used, it would be valuable to show the prior network statistics including the average degree, degree distribution, number of nodes, etc. Similarly it would be helpful to show these properties of the inferred networks.

5. The architecture of the CEFCON framework could be better motivated. Although it is explained that the two layers consider the embeddings when a gene is either a source or a target, it might be worth considering what happens if this was encapsulated in a single layer but still using the directionality to define the neighbors.

6. Figure 3 reports Early Precision which is a fraction of the true positives to the top k predicted edges. It is not clear why the number is greater than 1. The authors also need to explain Supp Fig 2. It is not clear what is accumulated AUPR or EPR which is used for the overall comparison. Please also explain how the error bars were computed (what is different between the 20 repeats).

7. The authors reanalyzed the hematopoietic system dataset to obtain cells in the erythroid, granulocyte-monocyte, and lymphoid lineages. However, they did not compare these newly inferred networks to those that were available as part of the benchmark. It would be very beneficial to understand how the reanalysis of the data affected the quality of the inferred edges.

8. The notion of precision with regards to the GO terms in figure 5 is misleading. It is simply if regulon has at least one enriched term. Furthermore number of enriched terms may not necessarily translate to more biologically meaningful gene sets.

9. It's great that the authors compared CEFCON-based prioritization to other methods. However, the comparison again is not quite right because it is not clear whether it is CEFCON's GRN that is helping to get better regulators or is it the specific metrics they used (MFVS and MDS, which is not novel). A more informative comparison would be if they compared MFVS and MDS on SCENIC's network and also on the input network since it had good performance on several benchmarks.

Minor:

1. The graph attention framework has been used to infer regulatory interactions, though in an unsupervised setting. The authors need to cite this <https://academic.oup.com/bioinformatics/article/38/19/4522/6663989> and discuss it in the context of their work.

2. The authors use some adjectives both for their method and other methods, which are not necessary, e.g. "outstanding", or "impressive".

3. The MFVS and MDS metrics are interesting for finding important regulators. It would be helpful to show the comparisons to simpler node importance measures based on centrality

NCOMMS-23-05780-T

Response to Reviewers' comments

We are very grateful for the insightful comments and constructive suggestions from all the reviewers and the editor, which are helpful for improving the quality of our manuscript. We have carefully address the reviewers' concerns point by point. In particular, we have improved our manuscript through addressing the following major concerns raised by the reviewers:

1. We have conducted additional tests to further assess the reliability of CEFCON in both GRN construction and driver regulator identification. More specifically, we have included one additional baseline method for comparison in our evaluation of GRN construction and two more state-of-the-art baseline methods for evaluating driver regulator identification. Moreover, we have extended our benchmarking studies using four additional prior gene interaction networks to provide more comprehensive assessment of the effectiveness of CEFCON. In addition, regarding the identification of driver regulators, we have compared our strategy with those using traditional centrality metrics on the same GRNs constructed by CEFCON. Furthermore, we have conducted a comparative analysis of the required running time between CEFCON and three other baseline methods.
2. We have carried out more analyses about the novel findings uniquely identified by CEFCON in comparison to other methods.
3. We have carefully revised our manuscript to make our paper more readable and easier to understand. In particular, we have modified a number of inappropriate statements and improved the original descriptions of motivation and main conceptual advances of our work in the Introduction section. We have also discussed the impact of the completeness of prior gene interaction networks on GRN construction and driver regulator identification in the Discussion section. Finally, we have carefully proofread the entire manuscript to guarantee the accuracy and clarity of the language used.

We believe that our work has been substantially improved after incorporating all the valuable suggestions by the reviewers and the editor, and the revised manuscript should have met the publication standard of Nature Communications. Please find the detailed response to each of the concerns raised by the reviewers in the remaining part of this response. *The original comments are in black italics* and **our responses are marked in blue**. **The changes in the revised manuscript and the revised supplementary materials are highlighted in red.**

Reviewer #1 (Remarks to the Author):

In the manuscript entitled “A network-based framework for deciphering driver regulators of cell fate decisions from single-cell RNA-seq data”, Peizhuo et al. developed CEFCON, which employed a graph attention neural network under a contrastive learning framework to construct a cell-lineage-specific gene regulatory network (GRN). Firstly, the use of graph neural network with attention mechanism is creative and suitable, which is meaningful for considering the importance of genes. Secondly, the authors applied CEFCON to the data during the differentiation of mouse hematopoiesis, which could elucidate the lineage differences from the control theory. Thirdly, CEFCON could comprehensively construct GRNs from both TFs and non-TFs. Overall, CEFCON is a valuable framework to build GRN and find key genes, but some minor comments are still needed and listed below.

Response:

We thank the reviewer for the nice summary and insightful comments on our work. To address the reviewer’s concerns, we have conducted additional tests and analyses, and revised the manuscript accordingly. The point-to-point responses are provided below.

1.1 - CEFCON adopted a gene interaction network proposed in NicheNet as a prior network, which models intercellular communication by linking ligands to target genes. It will be more convincing if the authors can test GRN inferred from other tools.

Response:

We thank the reviewer for the insightful comments. As pointed out by the reviewer, the employed prior network from NicheNet was originally used for modelling intercellular communication. Nevertheless, since NicheNet provides a comprehensive and context-free gene interaction network through integrating over 50 publicly available data sources, using such an informative network as the prior gene interaction network in our CEFCON framework can offer a reliable and comprehensive resource for understanding cell-lineage-specific GRNs.

We also strongly agree the importance of evaluating the performance of CEFCON in GRN inference using different background gene networks. As suggested by the reviewer, we have expanded our computational tests by additionally incorporating four global and context-free gene interaction networks, including Harmonizome (Rouillard, et al. 2016), InWeb_InBioMap (Li, et al. 2017), PathwayCommons (Rodchenkov, et al. 2020) and Omnipath (Türei, et al. 2016), and then compared the resulting performance on GRN inference on both mESC and hESC datasets (Figure R1). In addition, following the suggestion of Reviewer #3 (Comment 3.3), we have also included a case of considering only directed edges within the NicheNet network in our comparison. The major statistical information of the above gene interaction networks has been provided in Supplementary Table 1 in the revised manuscript.

As shown in Figure R1(a) and (b), our method achieved the best performance in GRN inference when using the prior gene interaction network derived from NicheNet. We speculated that the completeness of the chosen prior gene interaction network can influence the performance of our method in GRN inference. Thus, we have carried out further analyses to quantitatively measure and compare the completeness of these prior gene interaction networks. Specifically, we calculated the difference between the number of edges in the prior gene interaction networks (i.e., $|E_{\text{prior_net}}|$) and the number of edges in the ground-truth network (i.e., $|E_{\text{ground_truth}}|$) (Figure R1(c)). A positive value of this difference indicates that the size of the prior gene interaction network exceeds that of the corresponding ground-truth network, while a negative value suggests that even if considering the entire prior gene interaction network, there are still some interactions in the corresponding ground-truth network that cannot be covered. Figure R1(c) shows that the prior gene interaction network from NicheNet emerged as the most complete network among all the compared networks, resulting in the best results for GRN inference. Please also refer to our response to Comment 3.1 of Reviewer #3 for further discussion regarding the influence of the completeness of the prior gene interactions on GRN inference.

In the revised manuscript, we have included these results to Supplementary Figure 18 and also added the relevant information in the Discussion and Methods sections (Line 403-406, Page 12 and Line 658-661, Page 21).

Figure R1. Performance evaluation on GRN construction using different prior gene interaction networks on both hESC and mESC datasets. a, The performance measured in terms of AUPRC. **b,** The performance measured in terms of EPR. The NicheNet network with only directed edges, denoted as NicheNet (directed only), and the other four baselines, including Harmonizome, InWeb_InBioMap, PathwayCommons, and Omnipath (interactions), were considered for comparison. **c,** The difference between the number of edges in the prior gene interaction networks (i.e., $|E_{\text{prior_net}}|$) and the number of edges in the ground-truth network (i.e., $|E_{\text{ground_truth}}|$).

References:

- Rouillard, Andrew D., et al. "The harmonizome: a collection of processed datasets gathered to serve and mine knowledge about genes and proteins." Database 2016 (2016).
- Li, Taibo, et al. "A scored human protein–protein interaction network to catalyze genomic interpretation." Nature methods 14.1 (2017): 61-64.
- Rodchenkov, Igor, et al. "Pathway Commons 2019 Update: integration, analysis and exploration of pathway data." Nucleic acids research 48. D1 (2020): D489-D497.
- Türei, Dénes, Tamás Korcsmáros, and Julio Saez-Rodriguez. "OmniPath: guidelines and gateway for literature-curated signaling pathway resources." Nature methods 13.12 (2016): 966-967.

1.2 - What is the computational efficiency of CEFCON?

Response:

We thank the reviewer for bringing up this practical question. We agree that computational efficiency is crucial and need to be declared in our manuscript. In particular, we first briefly discussed the time efficiency of our CEFCON method. Specifically, CEFCON primarily consists of two core components: GRN construction and driver regulator identification, which are based on graph attention neural networks and network control methods, respectively. The graph attention neural network is a computationally efficient deep learning model that can be easily parallelized across all nodes in the graph (Veličković, et al., 2018). Therefore, using GPUs for this calculation is recommended. For the network control-based methods (i.e., MDS and MFVS), we have implemented them using efficient graph reduction and integer linear programming (ILP) approaches. In particular, the ILP problems were solved using Gurobi (<https://www.gurobi.com/>), a highly efficient optimization solver. In conclusion, CEFCON is not a time-consuming method.

Furthermore, we have compared the required time of CEFCON with that of other three baselines, i.e., pySCENIC, CellRouter and CellOracle, on the hESC datasets with the number of genes varying from 500 to 5,000 (Figure R2). We chose these methods for comparison because they all used single-cell RNA-seq data to obtain key genes through GRN inference. This computational experiment was conducted on a 32-processor Linux server equipped with an AMD Ryzen Threadripper PRO 3975WX CPU (32 cores, 64 threads) and an NVIDIA A100 (40GB) GPU. We used the default parameters for all the compared methods. More specifically, CEFCON used a GPU for GRN construction and then a CPU for gene identification, while the other methods used CPU multi-core processing for computation. For each method, we calculated the sum of runtime on both parts, i.e., GRN construction and key gene identification. As shown in Figure R2, we found that CEFCON was relatively time-efficient compared to the baseline methods.

We have included these results to Supplementary Figure 17 and also added the relevant information in the Discussion section (Line 395-396, Page 12) of the revised manuscript.

Figure R2. The required time (in second) of different methods on the hESC dataset. The number of genes varied from 500 to 5,000. The time cost was the sum of the runtime of both GRN construction and key gene identification.

Reference:

Veličković, Cucurull, et al. " Graph Attention Networks." International Conference on Learning Representations 2018 (2018).

1.3 - There are some methods that were not included in the comparison, such as CellOracle (Nature, 2023) and some methods shown in the BEELINE framework. The authors are supposed to include the evaluation of these methods or give reasons for exclusion.

Response:

We thank the reviewer for the helpful suggestion on expanding our benchmarking tests. In the revised manuscript, we have added CellOracle (Kamimoto, et al., 2023) as a new baseline method for comparison in GRN construction (Figure R3). As shown in Figure R3, CEFCON still outperformed all the baseline methods, including CellOracle. For the concern about the comparison with the other methods from the BEELINE framework, we only selected GRNBoost2 in our original manuscript because it had demonstrated superior performance than the other compared methods, as shown in the BEELINE paper (Pratapa, et al., 2020). Moreover, the baseline DeepSEM (Shu et al., 2022) had demonstrated its superior performance over all the methods shown in the BEELINE framework according the original computational tests in the DeepSEM paper. Therefore, we did not compare the other methods from the BEELINE framework. The results on the expanded benchmarking tests on GRN construction are shown in Figure R3 and have also been included to Fig. 3 in the revised manuscript. We have also updated the related texts about these results in the Results section of our revised manuscript and added a brief introduction of CellOracle in the Section A.2 of Supplementary Notes.

Figure R3. Performance evaluation on cell-lineage-specific GRN construction. The performance of GRN construction on different benchmark scRNA-seq datasets, measured in terms of AUPRC (a) and EPR (b), respectively. Six baselines were considered for comparison, including SCINET, GRNBoost2, DeepSEM, NetREX, CellOracle and Random_NicheNet (i.e., randomly selecting edges from the prior gene interaction network).

References:

Kamimoto, Kenji, et al. "Dissecting cell identity via network inference and in silico gene perturbation." *Nature* 614.7949 (2023): 742-751.

Pratapa, Aditya, et al. "Benchmarking algorithms for gene regulatory network inference from single-cell transcriptomic data." *Nature methods* 17.2 (2020): 147-154.

Shu, Hantao, et al. "Modeling gene regulatory networks using neural network architectures." *Nature Computational Science* 1.7 (2021): 491-501.

1.4 - CEFCON performed best in terms of AUPRC (Fig. 3a), but the real values of AUPRC in all methods are low. And the benchmark results in BEELINE showed slightly higher scores of AUPRC. The authors should explain this or choose more suitable evaluation methods.

Response:

We thank the reviewer for bringing up this point. In fact, the BEELINE framework reported the AUPRC ratio (not the AUPRC), which is the AUPRC divided by that of a random predictor. The AUPRC ratio greater than one means that the performance is better than the random case. On the other hand, in our original manuscript, we directly used AUPRC for performance evaluation instead of the AUPRC ratio.

1.5 - The authors used three gene sets that are associated with cell fates as ground truth (Fig. 4a-4h), which seemed a little bit subjective, more evaluation could be included. And as said in the introduction, "it is generally challenging to distinguish whether these genes are marker or driver genes for controlling cell fates", the authors should further explain why CERCON identified are "driver genes" rather than "marker genes".

Response:

We appreciate the reviewer for raising this important point about further evaluating the capability of CEFCON in identifying driver genes associated with cell fates. We would like to emphasize that comprehensive assessment of computational approaches in identifying driver genes of cell fates remains challenging mainly due to the lack of databases containing ground-truth gene sets for various types of cell development and differentiation. Our study focused on embryonic stem cell development (also see Figure 4(a)-(h) in the original manuscript), and utilized three highly relevant gene sets from the Gene Ontology (GO) database as the ground-truths, namely 'cell fate commitment', 'stem cell population maintenance' and 'endoderm development'. These GO gene sets describe the primary features of ESC differentiation into endoderm cells, which thus can serve as important functional features for evaluating the performance of CEFCON in identifying driver genes in the related biological processes.

Besides the above GO gene sets, to provide a more robust evaluation, we also expanded our benchmarking tests to include additional experimentally validated genes curated from the previous researches (Kimber, et al. 2008; Young, et al., 2011). We considered these experimentally validated genes, which had been previously shown to control the state and

differentiation of embryonic stem cells, as valuable supplementary ground-truths in addition to the gene sets from the GO database. We believe that all the above gene sets collectively furnished a sufficient basis for objectively assessing CEFCON and baseline methods in the identification of crucial genes associated with cell fates. As shown in Figure 4(d) and (h) in our original manuscript, CEFCON consistently achieved superior performance over baselines in driver regulator identification across various datasets.

Additionally, we have analyzed the gene expression trends of the top-20 genes identified by CEFCON along the developmental pseudotime (Figure R4), to investigate whether these genes were correlated with this developmental process. As shown in Figure R4, these genes can be roughly categorized into two groups based on their gene expression trends (separated by dotted lines). One group displayed a significant decrease in expression during the developmental process, in which most of the genes were involved in maintaining embryonic stem cell pluripotency (Young, et al., 2011). While the other group exhibited a significant increase in expression, in which most of the genes contributed to cell differentiation into endoderm cells (D'Amour, et al., 2005). For a more detailed analysis of these genes and their biological significance, please also refer to our response to Comment 1.6 of this response. **We have added these results to Supplementary Figure 7 and also updated the relevant texts in the Results section (Line 235-236, Page 8) of the revised manuscript.**

In response to the reviewer's concern regarding the lack of clarity of "CEFCON can identify 'driver genes' rather than 'marker genes'", we would like to provide an explanation of the fundamental principle of our proposed method, which hopefully can help address the reviewer's concern. Specifically, we modeled the dynamics of the developmental process from a perspective of network-based control theory, which aims to find driver nodes controlling the state of the whole network, corresponding to the complex system of cell fates. It has been theoretically demonstrated that the methods, such as MFVS and MDS, for modeling non-linear systems can effectively identify the driver nodes controlling the states of the entire system (Zañudo, et al., 2017; Mochizuki, et al., 2013; Liu, et al., 2016). Therefore, from this point of view, we think that CEFCON can find the driver genes that play a crucial role in controlling cell fates, rather than just identifying marker genes. We apologize for any confusion or lack of clarity in our manuscript, and **have revised the relevant parts to provide a clearer description in the Introduction section (Line 55-58, Page 3).**

Figure R4. The gene expression trends along the developmental pseudotime of the hESC dataset. Gene expression trends were obtained using the scGTM toolkit (Cui, et al., 2022). The top-20 genes identified by CEFCON are shown. These genes can be roughly divided into two groups based on their gene trends (separated by a dotted line), with a significant decrease or increase along the developmental pseudotime.

References:

- Kimber, Susan J., et al. "Expression of genes involved in early cell fate decisions in human embryos and their regulation by growth factors." *Reproduction* 135.5 (2008): 635-647.
- Young, Richard A. "Control of the embryonic stem cell state." *Cell* 144.6 (2011): 940-954.
- D'Amour, Kevin A., et al. "Efficient differentiation of human embryonic stem cells to definitive

endoderm." *Nature biotechnology* 23.12 (2005): 1534-1541.

Zañudo, Jorge Gomez Tejeda, Gang Yang, and Réka Albert. "Structure-based control of complex networks with nonlinear dynamics." *Proceedings of the National Academy of Sciences* 114.28 (2017): 7234-7239.

Mochizuki, Atsushi, et al. "Dynamics and control at feedback vertex sets. II: A faithful monitor to determine the diversity of molecular activities in regulatory networks." *Journal of theoretical biology* 335 (2013): 130-146.

Liu, Yang-Yu, and Albert-László Barabási. "Control principles of complex systems." *Reviews of Modern Physics* 88.3 (2016): 035006.

Cui, Elvis Han, et al. "Single-cell generalized trend model (scGTM): a flexible and interpretable model of gene expression trend along cell pseudotime." *Bioinformatics* 38.16 (2022): 3927-3934.

1.6 - It will be more convincing if the authors can analyze novel interactions or driver genes that CEFCON predict while other framework failed.

Response:

We thank the reviewer for the valuable suggestion. We agree with the reviewer that it is necessary to showcase novel interactions or driver genes that can be uniquely identified by CEFCON compared to other frameworks.

In response to this feedback, we have additionally conducted a comparative analysis to highlight the driver regulators uniquely identified by CEFCON. More specifically, we compared the top-20 obtained genes from CEFCON with those obtained from other methods, including CellOracle, CellRouter and SCENIC, on the hESC dataset (Figure R5). As shown in Figure R5, among the 10 genes (highlighted in red) that were specifically detected by CEFCON, seven of them were non-TFs (marked with asterisks). Notably, all the 10 genes uniquely identified by CEFCON have been previously reported to directly affect or be associated with embryonic development (Figure R5(e)). We would like to emphasize that CEFCON is able to identify key non-TFs involved in cell fate decisions, while most of other approaches only find key TFs from the transcriptional regulatory networks. Although there have been many reports showing that TFs are crucial for regulating cell fates, it is undeniable that many non-TFs also play key roles in cell fate decisions (De Belly, et al., 2022). **We have added these results to Supplementary Figure 8 in the revised manuscript and also updated the relevant texts in the Results section (Line 236-238, Page 8).**

Moreover, we have carried out additional case studies to validate the interactions identified by CEFCON using the relevant ChIP-seq and epigenomic data. Specifically, we focused on the predicted gene interactions involving several identified driver regulators, namely *GATA6*, *GATA4*, *NANOG* and *CDH1* (Figure R6(a)), among which *GATA6* and *CDH1* were uniquely identified by CEFCON (Figure R5(a)). To validate these interactions, we utilized an experimental dataset from GEO (GSE:213394), which comprised the ATAC-seq, H3K27ac and *GATA6* ChIP-seq data at seven time points during the differentiation of

hESCs to definitive endoderm (DE) cells. In addition, we obtained the *NANOG* ChIP-seq data of the H1 cell line and the H3K4me3 data of endodermal cells from the ENCODE database (Figure R6(b-c)). As shown in Figure R6, both *CDH1* and *GATA4* displayed significant distinct H3K4me3, H3K27ac and ATAC-seq peaks in the promoter regions, indicating strong transcriptional activity of these two genes. Meanwhile, the presence of the *NANOG* ChIP-seq peak at the *CDH1* promoter and the *GATA6* ChIP-seq peak at the *GATA4* promoter suggested that *NANOG* might regulate *CDH1* and *GATA6* might regulate *GATA4*. In fact, it had been previously reported that these regulatory relationships play key roles in the maintenance of stem cell pluripotency and the activation of the development towards the endoderm lineage (Jackson, et al., 2016; Fisher, et al., 2017). Furthermore, the substantial overlap between the *GATA6* and *NANOG* peaks at the *GATA4* promoter was consistent with a previous study, which reported that *GATA6* and *NANOG* co-bind at the vast majority of the same regulatory elements. This co-binding behavior maintains embryoblast plasticity and thus allows them to control lineage specification into either the primitive endoderm or epiblast (Thompson et al., 2022). Overall, these results provided a strong and orthogonal evidence to support the interactions predicted by CEFCON between these driver regulators. In the revised manuscript, we have added these results to Supplementary Figure 10 and also added the corresponding analyses in the Results section (Line 248-251, Page 8).

References:

- De Belly, Henry, Ewa K. Paluch, and Kevin J. Chalut. "Interplay between mechanics and signalling in regulating cell fate." *Nature Reviews Molecular Cell Biology* 23.7 (2022): 465-480.
- Jackson, Steven A., et al. "Alternative routes to induced pluripotent stem cells revealed by reprogramming of the neural lineage." *Stem Cell Reports* 6.3 (2016): 302-311.
- Fisher, J. B., et al. "GATA6 is essential for endoderm formation from human pluripotent stem cells." *Biology Open* 6.7 (2017): 1084-1095.
- Thompson, Joyce J., et al. "Extensive co-binding and rapid redistribution of NANOG and GATA6 during emergence of divergent lineages." *Nature Communications* 13.1 (2022): 4257.
- Robinson, James T., et al. "Integrative genomics viewer." *Nature biotechnology* 29.1 (2011): 24-26.
- ENCODE Project Consortium. "An integrated encyclopedia of DNA elements in the human genome." *Nature* 489.7414 (2012): 57.

Gene	Description	References
NOG	Noggin (NOG) regulates embryonic development by inhibiting BMP (bone morphogenetic proteins) signaling.	[21], [22]
CYP26A1	CYP26A1 regulates retinoic acid levels, thereby affecting the differentiation process in endodermal cells.	[23], [24]
GATA6	GATA6 defines endoderm fate by controlling chromatin accessibility during differentiation of pluripotent stem cells.	[25], [26]
LHX1	LHX1 is involved in epiblast development in embryonic stem cells.	[27], [28]
PITX2	The PITX2 homeobox protein plays an essential role early in the nodal signaling pathway, directing the specification of both endodermal and mesodermal germ layers.	[29], [30]
CDH1	E-cadherin (CDH1) plays a vital role in maintaining cell–cell adhesions in embryonic stem cells (ESCs), which is crucial for embryonic stem cell pluripotency.	[31], [32]
VIM	Vimentin (VIM) is often used as a marker of mesenchymally derived cells or cells undergoing an epithelial-to-mesenchymal transition (EMT) during normal embryonic development.	[33]
EPCAM	EPCAM is a surface marker on undifferentiated hESCs and plays functional roles in proliferation and differentiation.	[34], [35]
ERBB4	ERBB4, which is a receptor tyrosine kinase, is linked to canonical and noncanonical Wnt signaling which can determine the cellular fate of human pluripotent stem cells.	[36], [37]
HAPLN1	HAPLN1 is a major component of the ECM (extracellular matrix), which is of great significance to tissue development (e.g., developmental hematopoiesis).	[38]

Figure R5. The top-20 genes identified by different methods, including CEFCON (a), CellOracle (b), CellRouter (c) and SCENIC (d), on the hESC dataset. Genes uniquely detected by CEFCON are highlighted in red, and non-TFs are marked with an asterisk. e, Descriptions of the genes uniquely identified by CEFCON among the top-20 predicted genes and the related literatures.

Figure R6. Gene interactions between *NANOG*, *GATA6*, *CDH1* and *GATA4* predicted by CEFCON on the hESC dataset and their validation using the related ChIP-seq and epigenetic data. a, An illustration on the interactions between *NANOG*, *GATA6*, *CDH1* and *GATA4*. **b-c**, The *GATA6* and *NANOG* ChIP-seq, ATAC-seq, H3K27ac and H3K4me3 peaks at promoters and the gene bodies of *CDH1* (**b**) and *GATA4* (**c**) from the development of embryonic stem cells to definitive endoderm cells, visualized with the IGV genome browser (Robinson, et al., 2011). The ATAC-seq, H3K27ac and *GATA6* ChIP-seq data at 48-hour post differentiation were obtained from GEO (GSE213394). The H3K4me3 and *NANOG* ChIP-seq data were obtained from the ENCODE database (ENCODE Project Consortium, 2012). Shaded regions indicate the significant peaks in promoter regions of the target genes.

1.7 - CEFCON constructs GRNs by focusing on transcriptional regulation and signal transduction from both TFs and non-TFs, which is very comprehensive. I was wondering if TFs have more interaction than non-TFs in the network CEFCON inferred. And the author could give examples of several regulation patterns that do not require TF.

Response:

We thank the reviewer for the thoughtful comments. In response to the question regarding the relative interaction frequencies between individual TFs and non-TFs within the network inferred by CEFCON, we have conducted an analysis to calculate the average in/out-

degree of both TFs and non-TFs within the CEFCON-inferred networks (Table R1). Our analysis revealed that individual TFs indeed exhibited a higher number of interactions compared to non-TFs in the network, particularly in terms of their out-degrees. This observation was consistent with the well-established knowledge in the field that TFs play critical roles in cell fate decisions and tend to have extensive regulatory interactions (Gerstein, et al., 2012). **To improve the clarity of this point, in the revised manuscript, we have added Table R1 to the supplementary materials (Supplementary Table 4), and also added the relevant discussions in the Results section (Line 187-191, Page 6).**

Next, as suggested by the reviewer, we discuss a few examples of regulatory mechanisms that do not directly involve TFs. In addition to the direct TF-target relationships, various other mechanisms, including post-transcriptional modifications, transcriptional co-factors and epigenetic regulations, can also significantly influence gene expression and thereby impact the cell fate decisions (Badia-i-Mompel, et al., 2023). Many of these non-TF-mediated regulation patterns can be represented by protein-protein interactions, which can regulate the activity or localization of proteins. For example, the SWI/SNF (Switch/Sucrose Non-Fermentable) complexes and other chromatin remodeling complexes alter nucleosome positioning and chromatin accessibility, thus affecting the gene expression patterns, even though they are not TFs themselves (Clapier, et al. 2009). Another example is transcriptional co-factors, such as the retinoblastoma protein (*RB*), which acts as an adaptor protein that can bind to over one hundred protein partners, mediating transcriptional regulation of hundreds of target genes (Burkhart, et al., 2008). Furthermore, signaling pathways like the MAPK (Mitogen-Activated Protein Kinase) pathway rely on phosphorylation to regulate protein activity without TFs, in which the protein kinases are responsible for the mechanism of phosphorylation, activating or inhibiting proteins like TFs through adding or removing phosphate groups (Ardito, et al. 2017).

Table R1. The in-degree and out-degree statistics of TFs and non-TFs of the GRN constructed by CEFCON for each scRNA-seq dataset used in this study. The top 1,000 highly-variable genes were considered for GRN construction.

Dataset	#TFs	#non-TFs	Avg. in-degree of TFs	Avg. out-degree of TFs	Avg. in-degree of non-TFs	Avg. out-degree of non-TFs
hESC	86	819	9.4070	52.7326	3.4652	8.0147
hHEP	49	772	6.3061	29.6939	8.1334	9.6179
mESC	104	773	10.2788	29.6346	5.3777	7.9819
mDC	51	776	11.5882	65.4510	4.3015	7.8415
mHSC-E	58	846	10.5000	71.2931	3.7518	7.9196
mHSC-GM	66	822	8.5606	43.6515	5.2384	8.0560
mHSC-L	74	797	8.5541	48.5000	4.2447	7.9536

References:

- Gerstein, Mark B., et al. "Architecture of the human regulatory network derived from ENCODE data." *Nature* 489.7414 (2012): 91-100.
- Badia-i-Mompel, Pau, et al. "Gene regulatory network inference in the era of single-cell multi-omics." *Nature Reviews Genetics* (2023): 1-16.

Clapier, Cedric R., and Bradley R. Cairns. "The biology of chromatin remodeling complexes." *Annual review of biochemistry* 78 (2009): 273-304.

Burkhart, Deborah L., and Julien Sage. "Cellular mechanisms of tumour suppression by the retinoblastoma gene." *Nature Reviews Cancer* 8.9 (2008): 671-682.

Ardito, Fatima, et al. "The crucial role of protein phosphorylation in cell signaling and its use as targeted therapy." *International journal of molecular medicine* 40.2 (2017): 271-280.

Reviewer #2 (Remarks to the Author):

Wang et al. described an algorithm to infer gene regulatory networks (GRNs) by integrating prior protein interaction data, single-cell RNA-sequencing (scRNA-seq) and a deep learning framework based on graph convolutional networks with attention mechanisms. The algorithm, CEFCON, predicts cell or state-specific GRNs based on contrastive learning, and applies control theory to rank critical regulatory nodes, as well as their regulons based on the inferred GRNs. This approach is flexible with regards to the identification of regulators that are transcription factors (TFs) and non-TFs. The resulting regulons capture expected biological processes. The authors applied CEFCON to scRNA-seq datasets profiling embryonic stem cells and hematopoietic cell types, and found regulators and biological processes previously reported in the literature. Overall, the methodology is interesting and by comparing predicted cell or state-specific GRNs to ChipSeq gold standards of the respective cell lineages, the authors show that CEFCON outperforms competing methods in terms of AUPRC. However, I have some concerns:

Response:

We sincerely thank the reviewer for the nice summary and valuable comments on our work, which have greatly helped improve the quality of our manuscript. Below we will provide our point-to-point responses to the reviewer's comments.

2.1 - It is not clear how CEFCON translates into novel biological insights. The authors seem to only focus on known regulators or biological processes, which provides some confidence that the method works, but do not explain any potential new insights or gene programs/regulons that could be uniquely identified by CEFCON and NOT identified by other methods, which then would translate to new biological insights, specially in the presence of experimental follow-up.

Response:

We thank the reviewer for raising this important point. We strongly agree with the reviewer that it is essential to demonstrate the capability of CEFCON in identifying potential novel regulators. We thus additionally cross-checked our results using the previously reported findings from existing literatures as well as the data from other known experiments.

In particular, in response to this point, we have conducted additional analysis on the driver regulators that were uniquely identified by CEFCON but could not be obtained by other existing frameworks. Specifically, we compared the top-20 genes identified from CEFCON with those obtained from other methods, including CellOracle, CellRouter and SCENIC, on the hESC dataset (Figure R7). We chose these methods for comparison because they all identify key genes through constructing GRNs, which is similar to the strategy applied in CEFCON. As shown in Figure R7, 10 genes (highlighted in red) were specifically detected by CEFCON, and seven of them were non-TFs (marked with asterisks). Importantly, all the genes that were uniquely identified by CEFCON have been reported to directly influence

or be associated with embryonic development (Figure R7(e)). This observation suggested that CEFCON was able to identify crucial non-TFs involved in cell fate decisions, while most of the benchmarked methods only focused on identifying key TFs from the transcriptional regulatory networks (e.g., CellOracle and SCENIC). Although numerous studies have emphasized the significance of TFs in controlling cell fates, it is undeniable that certain non-TFs can also play substantial roles in controlling cell fate decisions (De Belly, et al., 2022). **We have added these results to Supplementary Figure 8 in the revised manuscript and also updated the texts in the Results section (Line 236-238, Page 8).**

Furthermore, in our analysis of mouse hematopoietic stem cells differentiation in the Results section of the original manuscript, CEFCON identified *Dusp1* as a driver regulator in all three developmental lineages of mouse hematopoietic stem cells, i.e., erythroid lineage, granulocyte-monocyte lineage, and lymphoid lineage. We also observed that *Dusp1* was highly expressed in hematopoietic stem cells and showed a decreasing trend in expression in the differentiation progresses (please also see Figure 5(d) in our original manuscript). Notably, *Dusp1* has been proven to be a proliferation-associated gene in previous research (Fan, et al., 2021), while it is not a TF gene and thus cannot be identified by other TF-based methods. These findings suggested that *Dusp1* can play a pivotal role as a regulator in controlling the differentiation of hematopoietic stem cells, thus demonstrating the potential of CEFCON in identifying novel driver genes including both TFs and non-TFs associated with cell fates

To sum up, based on the known experimental results that have been previously reported, we believe that CEFCON has the great potential in discovering novel interactions or driver genes critical to cell fate decisions. We would also like to emphasize that the extensive benchmarking tests conducted in our manuscript clearly illustrated the superior accuracy and stability of CEFCON in GRN construction and driver gene identification in comparison with all the baseline methods. All these observations collectively demonstrated the potential of CEFCON in identifying novel regulators in important biological processes, and thus providing biologically useful insights.

References:

Gerstein, Mark B., et al. "Architecture of the human regulatory network derived from ENCODE data." *Nature* 489.7414 (2012): 91-100.

De Belly, Henry, Ewa K. Paluch, and Kevin J. Chalut. "Interplay between mechanics and signalling in regulating cell fate." *Nature Reviews Molecular Cell Biology* 23.7 (2022): 465-480.

Gene	Description	References
NOG	Noggin (NOG) regulates embryonic development by inhibiting BMP (bone morphogenetic proteins) signaling.	[21], [22]
CYP26A1	CYP26A1 regulates retinoic acid levels, thereby affecting the differentiation process in endodermal cells.	[23], [24]
GATA6	GATA6 defines endoderm fate by controlling chromatin accessibility during differentiation of pluripotent stem cells.	[25], [26]
LHX1	LHX1 is involved in epiblast development in embryonic stem cells.	[27], [28]
PITX2	The PITX2 homeobox protein plays an essential role early in the nodal signaling pathway, directing the specification of both endodermal and mesodermal germ layers.	[29], [30]
CDH1	E-cadherin (CDH1) plays a vital role in maintaining cell–cell adhesions in embryonic stem cells (ESCs), which is crucial for embryonic stem cell pluripotency.	[31], [32]
VIM	Vimentin (VIM) is often used as a marker of mesenchymally derived cells or cells undergoing an epithelial-to-mesenchymal transition (EMT) during normal embryonic development.	[33]
EPCAM	EPCAM is a surface marker on undifferentiated hESCs and plays functional roles in proliferation and differentiation.	[34], [35]
ERBB4	ERBB4, which is a receptor tyrosine kinase, is linked to canonical and noncanonical Wnt signaling which can determine the cellular fate of human pluripotent stem cells.	[36], [37]
HAPLN1	HAPLN1 is a major component of the ECM (extracellular matrix), which is of great significance to tissue development (e.g., developmental hematopoiesis).	[38]

Figure R7. The top 20 genes identified by different methods, including CEFCON (a), CellOracle (b), CellRouter (c), and SCENIC (d), on the hESC dataset. Genes uniquely detected by CEFCON are highlighted in red, and each non-TF is marked with an asterisk. e, Descriptions and related literature of the genes uniquely identified by CEFCON among the top-20 predicted genes.

2.2 - The high computational demand needed to run CEFCON, which according to their github page, ideally requires processing in graphics processing units (GPU), warrants the authors to better describe how their method could identify new regulators and their regulons (therefore, new biology) when compared to other methods. If CEFCON does contribute to identifying novel biology relative to other methods (which typically demand lower computational resources). In the absence of these comparisons, although the approach taken by the authors is interesting, it does not seem that CEFCON represents a substantial advance over prior methods, especially because there are no experimental follow-ups of CEFCON predictions and their implications for cell differentiation.

Response:

We appreciate the reviewer's comments regarding the trade-off between the computational

demand and methodology contributions. In our original Github page, we provided a rough estimate of the computational time cost. In response to this concern, we have conducted a more comparative analysis of the required time for CEFCON and three other baselines, i.e., pySCENIC, CellRouter, and CellOracle, through testing them on the hESC datasets with the number of genes varying from 500 to 5,000 (Figure R8). We chose these methods for comparison because they all used single-cell RNA-seq data to obtain key genes through GRN inference, which was similar to our approach. Our computational experiments were conducted on a 32-processor Linux server with an AMD Ryzen Threadripper PRO 3975WX CPU (with 32 cores and 64 threads) alongside an NVIDIA A100 (40GB) GPU. All the compared methods were evaluated under their default settings. More specifically, CEFCON employed the GPU for constructing the GRN and the CPU for identifying driver regulators, while the other approaches used multi-core CPU processing for their computation. For each approach, we calculated the sum of the running time of two parts, i.e., GRN construction and key gene identification. As illustrated in Figure R8, the tests showed that CEFCON was relatively time-efficient compared to the baseline methods. **We have added these results to Supplementary Figure 17 and also added the relevant information in the Discussion section of our revised manuscript (Line 395-396, Page 12).**

In response to the reviewer's concern about the computational resource requirement of our method, it's worth noting that some baseline methods, such as CellOracle and SCENIC, while only requiring CPUs for computation, actually benefit from multi-core processing to achieve high efficiency. For our approach, we recommend employing GPUs for GRN construction, while the subsequent driver gene identification process is highly time-efficient on a single CPU mainly due to the use of efficient graph reduction strategies and an efficient optimization solver. We would like to point out that compared to the performance improvement, such time cost is relatively acceptable for deep learning-based methods. Moreover, with the widespread applications of deep learning methods in recent years, the availability of GPU resources has become increasingly common.

Regarding the reviewer's comment about the lack of experimental follow-ups in this study, as addressed in our response to Comment 2.1, we have conducted an additional analysis to highlight the novel driver regulators uniquely identified by CEFCON compared with other baseline methods. This result further provided a strong evidence to support the unique strength of our method over the baseline approaches.

Figure R8. The required time (in second) of different methods on the hESC dataset. The number of genes varied from 500 to 5,000. The time cost was the sum of the running time of both GRN construction and key gene identification.

2.3 - In the introduction, lines 19-22, the authors indicate that algorithms to infer pseudo-temporal order of cells from scRNA-seq neglect how intracellular factors control cell fate changes. I believe this is not totally true as currently available algorithms directly or indirectly identify such intracellular factors controlling cellular differentiation. Once a developmental order is identified, virtually all papers aim to identify gene programs controlling these transitions, although many do not necessarily include a GRN inference step. For instance, Lummertz da Rocha et al, Nature Communications 2018, explicitly reconstructs gene regulatory networks directly from scRNA-seq data, and establishes a score that integrates the GRN with the kinetic profiles of genes during the inferred trajectory, prioritizing transcriptional regulators (not only transcription factors) that modulate cell fate decisions. Another example is the work by Alsinet et al, Nature Communications, 2022, which uses scVelo to identify transcriptional regulators involved in myelopoiesis. There are many other examples and strategies to identify genes that control cell fate decisions. Therefore, this seems an overstatement and authors should focus on what their method is designed to do (GRN inference), instead of what is a downstream application (trajectory inference) of what they aim to achieve with their work.

Response:

We greatly appreciate the reviewer for the comments with specific examples of existing methods that consider intracellular factors in controlling cellular differentiation. We agree that currently available methods can directly or indirectly identify intracellular factors that control cell differentiation, and our description about this point in the original manuscript was somewhat overstated. According to the reviewer’s suggestion, we have deleted the inappropriate statement in lines 19-22 of the original manuscript and modified the corresponding statement in the Introduction section to more precisely describe our motivation, that is,

“Despite these advancements, comprehending the mechanisms of how cells are controlled to determine their fates still remains a challenge.” (Line 21-22 in the revised manuscript)

Furthermore, we would like to emphasize that CEFCON focuses on identifying driver regulators that control cell fate decisions through constructing cell-lineage-specific GRNs. To enhance the clarity, we have also updated and added the corresponding descriptions in the Introduction section, that is,

“Nonetheless, there still remains a significant need for methods that can directly elucidate the driver roles of these genes in controlling cell fates.” (Line 34-35 in the revised manuscript)

...

“Through control theory, the complex systems within cells can be modeled as GRNs, where driver nodes are defined as those critical genes that drive the entire system to a desired state through perturbation [24]. Thus, control theory may offer useful insights into the identification of driver regulators controlling cell fate decisions.” (Line 55-58 in the revised manuscript)

In addition, we have carefully surveyed the examples of existing methods provided by the reviewer as well as other related ones, and additionally selected typical methods, including CellOracle (Kamimoto, et al., 2023) and CellRouter (Lummertz, et al., 2018), as additional baselines for our comparison in both GRN construction and driver gene identification tasks. The results demonstrated that CEFCON still performed the best in GRN construction (Figure R9) and can identify driver regulators more accuracy than most of other methods (Figure R10). We have included these results to Figure 3 and Figure 4, and also updated the relevant information in the Results section of the revised manuscript.

Figure R9. Performance evaluation on cell-lineage-specific GRN construction. The performance of GRN construction on different benchmark scRNA-seq datasets, measured in terms of AUPRC (a) and EPR (b), respectively. Six baselines were considered for comparison, including SCINET, GRNBoost2, DeepSEM, NetREX, CellOracle and Random_NicheNet (i.e., randomly selecting edges from the prior gene interaction network). The mean \pm standard deviation scores over 20 repeats were reported.

Figure R10. Performance evaluation on driver regulator identification on the mESC (a-d) and hESC (e-h) datasets. The performance was measured in terms of the precision of the top- k predicted genes among all known genes in the four ground-truth gene sets. Five baselines were considered for comparison, including SCENIC, VIPER, ANANSE, CellOracle and CellRouter. All the results with k ranking from 1 to 20 were reported. The shaded area represents the variation of precision over 20 repeats.

References:

- Kamimoto, Kenji, et al. "Dissecting cell identity via network inference and in silico gene perturbation." *Nature* 614.7949 (2023): 742-751.
- Lummertz da Rocha, Edroaldo, et al. "Reconstruction of complex single-cell trajectories using CellRouter." *Nature communications* 9.1 (2018): 892.

2.4 - Although CEFCON is based on prior protein and regulatory interaction data to infer GRNs, which likely reduces false-positive interactions, it also implies that new regulatory interactions might not be uncovered, when compared to that possibility when using de novo GRN inference methods (which, of course, have a higher false-positive rate). It is not surprising that CEFCON performs better in terms of AUPRC since it is based on prior gene/protein interaction network data. The fact that CEFCON needs to be applied

separately for each cell type or state present in the dataset makes the analysis workflow cumbersome.

Response:

We thank the reviewer for providing valuable insights regarding the trade-off and consideration associated with CEFCON's reliance on prior interaction data, its performance, and its practicality for multiple cell types or states.

Regarding the reviewer's concern about potentially missing new interactions when using our method compared to the *de novo* GRN inference methods, we implemented the following two strategies in our original manuscript to minimize the absence of meaningful interactions. First, it is crucial to select a comprehensive and high-quality prior gene interaction network that covers as many biologically significant gene interactions as possible. In our CEFCON framework, we utilized the gene interaction network from NicheNet, which integrated a large number of protein-protein interactions, transcriptional regulatory relationships, signaling pathways, and other interactions from over 50 data sources. In the revised manuscript, we have further conducted more comprehensive evaluation tests of CEFCON on GRN inference and driver regulator identification using additional global gene interaction networks as the input prior network, including Harmonizome (Rouillard, et al. 2016), InWeb_InBioMap (Li, et al. 2017), PathwayCommons (Rodchenkov, et al. 2020) and Omnipath (Türei, et al. 2016)) (please see Supplementary Table 1 in the revised manuscript for the basic statistics of these networks). The new evaluation results on both GRN construction (Figure R11) and driver regulator identification (Figure R12) demonstrated that the best results were obtained through using the NicheNet network, suggesting that its abundant genes and interactions can help our model obtain more reliable results. Second, for each specific dataset, we also calculated the gene co-expression relationships using Spearman's correlation coefficients based on the input gene expression profiles, and then supplemented the prior gene interaction network by incorporating the highly correlated relationships that were not originally present in the network. In conclusion, by considering the above strategies, we believe that CEFCON can yield a more reliable GRN and reduce the absence of meaningful interactions. **In the revised manuscript, we have included these results in Supplementary Figure 18 and Supplementary Figure 19, and the corresponding discussion in the Discussion section (Line 400-407, Page 12) of the revised manuscript.**

Furthermore, as shown in Figure R11 in our response to Comment 2.3, CEFCON still achieved higher AUPRC scores for GRN construction compared to other prior-based methods (e.g., SCINET and NetREX) using the same input prior network.

Moreover, we would like to emphasize that our network control-based strategy for identifying driver regulators associated with cell fates is highly dependent on the reliable GRNs with low false-positive rates. Hence, we tend to reduce the false positives to ensure the reliability of our predicted interactions.

In response to the reviewer's concern that CEFCON need to be applied separately for each cell type or state, we have updated the code on our Github (<https://github.com/WPZgithub/CEFCON>) to enable CEFCON to perform computation for each cell lineage/state automatically, thus making it more user-friendly.

Figure R11. Performance evaluation on GRN construction using different input prior gene interaction networks on the hESC and mESC datasets. a, Performance measured in terms of AUPRC. **b,** Performance measured in terms of EPR. The NicheNet network with only directed edges, i.e., NicheNet (directed only), and the other four baselines, including Harmonizome, InWeb_InBioMap, PathwayCommons, and Omnipath (interactions), were considered for comparison.

Figure R12. Performance evaluation on driver regulator identification using different input prior gene interaction networks. a, Performance evaluation on the hESC dataset. **b,** Performance evaluation on the mESC datasets. The NicheNet network with only directed edges, i.e., NicheNet (directed only), and the other four baselines, including Harmonizome, InWeb_InBioMap, PathwayCommons, and Omnipath (interactions), were considered for comparison. The precision scores of the top-10 and top-20 predicted genes among all known genes in the four ground-truth gene sets were calculated, respectively.

Minor comments:

2.5 - It is not clear how node centrality was calculated

Response:

We thank the reviewer for this feedback. The node centrality in our CEFCON framework, i.e., the influence score, was calculated based on the attention coefficients obtained from the constructed GRN. More specifically, we define the influence score of a gene as the log-transformed sum of the scaled attention coefficients of the gene's neighbors. We first calculate the influence score for each gene on the out-going and in-coming networks individually, and then linearly combine them together, that is:

$$\begin{aligned}
S_i^{in} &= \ln(1 + \sum_{j \in \mathcal{N}_i^p} \beta_{ji}^{out}), \\
S_i^{out} &= \ln(1 + \sum_{j \in \mathcal{N}_i^s} \beta_{ij}^{in}), \\
S_i &= \lambda S_i^{out} + (1 - \lambda) S_i^{in},
\end{aligned}$$

where β_{ji}^{out} and β_{ij}^{in} are the scaled attention coefficients obtained from the out-going and in-coming networks, respectively, \mathcal{N}_i^p and \mathcal{N}_i^s stand for the sets of predecessor and successor neighbors of node v_i , respectively, and $\lambda \in [0,1]$ stands for a parameter balancing these two terms. Detailed calculation can be found in Section 4.2.3 of our manuscript.

In our CEFCON framework, we first obtained the candidate driver gene candidates based on the network control methods (i.e., MDS and MFVS), and then ranked the candidate driver genes based on the above defined influence scores, to obtain the important driver regulators.

2.6 - It is not clear what are the conceptual differences between CEFCON, DeepSem, and competing methods.

Response:

We thank the reviewer for this comment. We would like to emphasize that our CEFCON method is designed to identify driver regulators that control cell fate decisions through constructing cell-lineage-specific GRNs. Among the baseline methods, only CellOracle, SCENIC and CellRouter have similar goals with our method, i.e., identifying key genes through GRN construction. To facilitate a better understanding of the compared methods, we have provided a brief introduction to all the methods in Supplementary Note A.2 of our manuscript.

Regarding the difference between CEFCON and another deep learning based approach DeepSEM, they mainly differ in the following two aspects: 1) DeepSEM is mainly designed for GRN construction and can be used for cell clustering and embedding. In contrast, CEFCON not only constructs cell-lineage-specific GRNs, but also identifies driver regulators that control cell fate changes based on the constructed GRNs; 2) DeepSEM is a deep generative model without relying on any prior knowledge, whereas CEFCON is a graph-based method utilizing graph attention neural networks for GRN construction, thus requiring a prior gene interaction network as the background network.

2.7 - It is not clear how directionality of message-passing (Line 90) gene-gene interactions was defined to create out-degree and in-degree types of networks used during training and subsequently to identify regulatory interactions? Was this information available in the prior

interaction data?

Response:

We thank the reviewer for highlighting the need for further clarifying the definition of the directionality of both out-going and in-coming networks. In the CEFCON framework, the directionality of message-passing within GNNs was determined mainly based on the direction of the edges in the underlying network. Specifically, as illustrated in Figure R13, if the message-passing direction is consistent with the direction of edges, i.e., flowing from a source node to a target node, it is defined as the in-coming network. Conversely, the direction of message-passing in the out-going network is from a target node to a source node. The resulting feature embeddings of the both in-coming and out-going networks are concatenated and encapsulated in a single layer, and then passed into a subsequent GNN layer. In fact, both out-going and in-coming networks share the same underlying network structure, with the only difference in the opposite directions of message-passing within the GNNs. The training process is conducted through deep graph infomax (DGI) (Veličković, et al., 2019), which is an unsupervised and contrastive learning framework. After training converges, the attention coefficients are then used to measure the strength of relationships between genes (see Section 4.1.5), and the top-ranked weighted edges are selected according to the average degree of the constructed GRN (see Section 4.1.6).

To make this statement clearer in our manuscript, we have modified Fig. 2(a) of the original manuscript by adding the message-passing directions in the illustration of the employed GNN model. We have also corrected and updated the relevant explanation (Line 90-93, Page 4) in the revised manuscript.

Figure R13. The gene regulatory network (GRN) construction module. a, A two-layer graph neural network (GNN) with multi-head attention mechanism is used for GRN construction in CEFCON. In each layer, the network structure is divided into in-coming and out-going networks based on the directions of message-passing to consider the importance of a gene as a regulator or target. The outputs of both directional networks are then concatenated together. **b,** A

contrastive learning model used to train the GNN encoder by maximizing the mutual information (MI) between the gene embeddings and their corresponding summary. The corruption operation produces the negative samples by randomly shuffling the node features. The differential gene expression level (log₂ fold change) is additionally considered as a learnable scalar encoding. After convergence, the attention coefficients are then used to construct the cell-lineage-specific GRN.

Reference:

Veličković, Petar, et al. "Deep graph infomax." International Conference on Learning Representations 2(3), 4 (2019).

2.8 - The author's tutorial in github should be improved to provide walk users through the analysis workflow, including output figures, how to interpret them, and how to possibly generalize the workflow for multiple cell types or states. The command line-only based approach is often not enough for users to confidently use the tool.

Response:

We thank the reviewer for this valuable feedback on our GitHub tutorial for the CEFCON method. We agree it is important to provide a more comprehensive and user-friendly analysis workflow for our tool. To address this concern, we have made substantial improvement to our code and updated the tutorials on GitHub (<https://github.com/WPZgithub/CEFCON>) to improve the user experience of utilizing CEFCON. Specifically, we have added new code for analyses and visualization, and provided the Jupyter notebooks for step-by-step guidance on various aspects, including input data preparation, generating cell-lineage-specific GRNs, identifying driver regulators, and conducting further analyses. We believe that these updates should be beneficial to users and help them utilize CEFCON for their research needs.

2.9 - Line 98: corresponding high-level summaries, WHICH ARE? Need to define what are input node features?

Response:

We apologies for the lack of clarity about these points. The 'corresponding high-level summaries' in line 98 of the original manuscript means the summary of all the node feature vectors. This concept comes from the adopted contrastive learning framework, i.e., DGI (Veličković, et al., 2019), where the summary vector of all the node feature representations can be used as a graph-level representation. DGI relies on maximizing the mutual information between the node feature representations and corresponding graph-level summaries. Compared to the node-level (i.e., local-level) representations, the graph-level representation is considered global- or high-level. **To enhance clarity and avoid any confusion, we have revised this sentence by replacing 'high-level summaries' with 'graph-level summaries (i.e., the summary vector of all the node feature representations)' in the Results section (Line 99-100, Page 4) of the revised manuscript.**

The initial input node features are gene expression profiles, and in each subsequent layer, the node features utilized are the embeddings learned from the previous layer of the neural network. More details about the input node features can be found in Section 4.1.7 of our manuscript.

Reference:

Veličković, Petar, et al. "Deep graph infomax." International Conference on Learning Representations 2(3), 4 (2019).

2.10 - Line 204: "the gaps between its performance and the best one were quite marginal." This is true overall for cases in which CEFCON provides slightly better predictions, or when other methods produce slightly better predictions. Therefore, unless the authors systematically determine what is uniquely identified by CEFCON and not by the other methods, the manuscript as it is does not represent a substantial advance over prior work.

Response:

We thank the reviewer for this comment. We agree with the reviewer that in the computational tests presented in Section 2.3 of our manuscript, CEFCON did not consistently yield the highest performance across all cases. We would like to emphasize that, except for accuracy, stability also stands as a significant characteristic of a reliable computational method. From our test results, CEFCON consistently achieved the top three performance in all cases. In contrast, all the baseline methods failed to consistently offer such stable performance. For example, SCENIC performed well on the hESC dataset, but exhibited significantly diminished performance on the mESC dataset (please see Figure R10 in our response to Comment 2.3). These observations demonstrated the superior stability of CEFCON compared to all the baseline methods. Overall, we believe that CEFCON can offer both accurate and robust prediction in driver regulator identification, thus making it a useful tool for deciphering cell fate decisions.

Regarding the unique findings of CEFCON compared to other methods, we have presented the advantage of CEFCON in identifying both important TFs and non-TFs in our response to Comment 2.1. Additionally, we have discussed how network control theory elucidated the 'driver' roles of the identified genes. Please see our response to Comment 2.1 for more details.

2.11 - Line 286-289: This is an interesting observation. What is Figure 5f heatmap showing though? How were the active and repressive genes determined? Do any of the genes identified as repressive or active in this section actually show this behavior experimentally in other papers? To be more specific: the genes themselves identified by CEFCON (and not by other methods) were experimentally tested to have such repressive or activation roles during differentiation?

Response:

We appreciate the reviewer's questions regarding Figure 5f in our manuscript. Below we will answer these questions one by one.

➤ *What is Figure 5f heatmap showing though?*

Figure 5(f) shows the AUCell activity heatmap of the identified out-degree types of regulon-like gene modules (RGMs) involving *PRDM1* for the hESC dataset. It was extracted from the heatmap shown in Figure 5(e). Since we did not find a clear activity pattern of the RGM involved in *PRDM1* during the developmental states in Figure 5(e), we further analyzed it in more detail, as shown in Figure 5(f). To be specific, we briefly calculated the co-expression between *PRDM1* and the member genes within its RGM, and divided those member genes into two groups based on the positive or negative correlations of co-expression. Subsequently, we calculated the AUCell activity matrices of these two groups and represented them in a heatmap, which is represented in Figure 5(f).

➤ *How were the active and repressive genes determined?*

To determine the active and repressive genes, we calculated the Pearson correlation coefficients between the expression profiles of *PRDM1* and the member genes within its RGM. The correlations greater than 0 indicated that the genes were activated by *PRDM1*, and conversely, those correlations less than 0 indicated that the genes were repressed. **To clarify this point, we have added a relevant description in Section 2.4 (Line 311-313, Page 10) of the revised manuscript.**

➤ *Do any of the genes identified as repressive or active in this section actually show this behavior experimentally in other papers? To be more specific: the genes themselves identified by CEFCON (and not by other methods) were experimentally tested to have such repressive or activation roles during differentiation?*

We would like to emphasize that the activation or repression here was defined according to their co-expression with *PRDM1*. Such a definition itself may not directly imply the functional roles of these genes during differentiation. Nevertheless, we observed that the *PRDM1*-repressed genes were mainly expressed at the early stage of embryonic stem cells, while the *PRDM1*-activated genes were expressed in the late stage towards endodermal cells (Figure 5f). Therefore, we could speculate that those two groups of genes may play different regulatory roles in the corresponding developmental stages.

In particular, we found that *PRDM1* was not actively expressed at the early stage of embryonic stem cells, while the expression increased significantly during differentiation towards endodermal cells (Figure R14(a)). Such an expression pattern of *PRDM1* was consistent with those of the *PRDM1*-activated genes, so it is natural to speculate that the *PRDM1*-activated genes may directly or indirectly play similar roles with *PRDM1*. As expected, the functional enrichment analysis of the *PRDM1*-activated genes on GO (biological process) terms also revealed that they were mainly associated with developmental functions (Figure R16(b)). In contrast, among the genes repressed by *PRDM1*, we identified a number of genes associated with stem cell pluripotency, such as

POLR3G (Lund, et al., 2017), *SFRP2* (Sato, et al., 2006), *FGF2* (Yu, et al., 2011) and *NANOG* (Wang, et al., 2012), among which *NANOG* was predicted to be the highest ranked driver regulator.

Figure R14. The gene expression trend of *PRDM1* over pseudotime (a) and the GO functional enrichment analysis of the genes activated by *PRDM1* within its RGM (b).

References:

Lund, Riikka J., et al. "RNA polymerase III subunit *POLR3G* regulates specific subsets of polyA+ and smallRNA transcriptomes and splicing in human pluripotent stem cells." *Stem Cell Reports* 8.5 (2017): 1442-1454.

Sato, Wataru, et al. "Sfrp1 and Sfrp2 regulate anteroposterior axis elongation and somite segmentation during mouse embryogenesis." (2006): 989-999.

Yu, Pengzhi, et al. "FGF2 sustains *NANOG* and switches the outcome of BMP4-induced human embryonic stem cell differentiation." *Cell stem cell* 8.3 (2011): 326-334.

Wang, Zheng, et al. "Distinct lineage specification roles for *NANOG*, *OCT4*, and *SOX2* in human embryonic stem cells." *Cell stem cell* 10.4 (2012): 440-454.

2.12 - Line 301: It would be interesting to see the CEFCON-repredicted HSC regulators.

Response:

We thank the reviewer for this helpful suggestion. In response, we have provided two additional sets of repredicted regulators related to mouse HSC differentiation under different random seeds (Figure R15). To further quantify the consistency of the results across these replicates, we calculated the overlap of identified driver regulators between each pair of replicated results for the erythroid, granulocyte-monocyte and lymphoid lineages, respectively (Figure R15(c)). These results showed that most driver regulators can be consistently identified by CEFCON, indicating the robustness of our method. In the revised manuscript, we have included these results in Supplementary Figure 14 and also updated the relevant texts (Line 328-329, Page 10).

Figure R15. Replicated results of the identified top-20 driver regulators for the three developmental lineages of mouse hematopoietic stem cell differentiation. These results were generated under different random seeds. The replicate 1 result was used for analyses in the main manuscript. **a**, The replicate 2 results. **b**, The replicate 3 results. **c**, The heatmaps illustrating the number of common genes between each pair of replicated results for the three developmental lineages.

2.13 - *The role of Gata2 and their kinetics during differentiation is overall well described during HSC differentiation. It feels like authors put too much emphasis on CEFCON-predictions that are already known, instead of systematically define what is unique to CEFCON.*

Response:

We thank the reviewer for this comment. Here, we would like to demonstrate that our method can yield reliable predictions which were consistent with the previously validated results. Our analyses of mouse hematopoietic stem cell differentiation revealed the driver regulators of the three lineages, i.e., erythroid lineage, granulocyte-monocyte lineage, and lymphoid lineage, among which *Gata2* was one of the top-ranked common driver regulators. Therefore, in our case study of regulon-like gene modules (RGMs), we used *Gata2* as an example for the detailed demonstration. In addition, we found that many of our identified driver regulators had been previously reported to be associated with hematopoietic stem cell differentiation, which thus supported our prediction results to some extent. We have added the corresponding references in Section 2.5 of the revised manuscript.

Overall, the authors developed an interesting algorithm based on graph neural networks to infer gene regulatory networks from single-cell RNA-sequencing data. However, it is not very clear from the paper what is the main conceptual advances of CEFCON, what new biology CEFCON can uncover despite having higher performance in terms of AUPRC, especially because improvements in performance relative to known gene sets were only marginally improved relative to other methods.

Response:

We appreciate the reviewer for the valuable feedbacks, which greatly help improve the clarity and significance of our work. We would like to emphasize that, in our CEFCON framework, the use of network control theory provides an intuitive description of GRN-based cell fate dynamics. Thus, in addition to achieving good performance, CEFCON has the natural potential to elucidate the "driver" roles of the identified regulators according to the network control theory, which we argue is one of the main conceptual advances of CEFCON. In addition, to fully address the reviewer's concern, we have also performed additional tests and analyses to further demonstrate that CEFCON was able to find unique and novel insights that could not be identified by other existing methods. Please refer to our response to Comment 2.1 for more details. We believe that our approach can provide a unique perspective in identifying gene regulatory networks and identifying driver regulators for controlling cell fate decisions from single-cell RNA-sequencing data, and thus provides a new powerful tool to the toolkit of single-cell data analyses.

Reviewer #3 (Remarks to the Author):

This paper presents CEFCON, a graph attention based method for inferring a lineage-specific gene regulatory network (GRN) from single cell RNA-seq data. The inferred network is analyzed using node prioritization methods based on network control theory to identify potential cell-fate drivers and is also analyzed to identify gene modules centered around the predicted drivers. CEFCON works by taking as input a non-specific graph, which is further augmented with co-expression edges and the cell line-specific scRNA-seq data as node/gene attributes and uses Deep Graph Infomax (DGI) to learn gene-level embeddings and attention coefficients. The attention coefficients are then used to infer edge weights which is used to either produce a ranking of edges in a lineage-specific manner or select the top $k_d \cdot N$ edges to obtain a network. CEFCON is evaluated on datasets available in the BEELINE benchmark dataset and was shown to have favorable performance compared to existing methods. The authors further demonstrate the driver regulator identification and gene module identification in the hematopoietic differentiation lineage. Overall, this is a clearly written paper with the main novelty being the use of graph attention to infer a context-specific network. However, the ability of CEFCON to truly model cell lineage dynamics seems limited and as such seems to be an approach that filters out edges that might be relevant in a particular condition. Accordingly the compared methods might not be most appropriate. There are also a few issues in terms of the input network, the architecture of the model and also evaluation of cell fate driver predictions that need further clarification. Though the GAT-based approach is interesting, the impact of this approach to infer cell lineage-specific gene regulatory networks seems limited. More detailed comments follow:

Response:

We thank the reviewer for the detailed summary and helpful suggestions on our manuscript. To address the reviewer's concerns, we have conducted several additional computational experiments and analyses, including further benchmarking against appropriate methods, tests on different prior gene interaction networks, clarification about the structure of our model, and more interpretation and analyses of the predicted driver genes. We have provided further clarification of these technical details in the following point-by-point response and revised the manuscript accordingly.

3.1 - The overall concept of CEFCON is to remove edges that are not supported by the input scRNA-seq dataset to infer what is referred to a lineage-specific network. This makes the final output depend upon the completeness of the input network, though they augment it with co-expression networks inferred from the same input scRNA-seq dataset. It would be important to show how much the expression versus the input network is playing a role and how much the completeness of the input network influences the results.

Response:

We appreciate the reviewer for raising this important point about the completeness of the

input network and its potential influence on the CEFCON results. To address this concern, we have conducted additional computational experiments to systematically evaluate the effect of using different input prior networks on GRN construction and driver regulator identification. Together with our previous results, we can gain useful insights into the respective roles played by gene expression data and input networks.

First, we tested CEFCON using additional gene interaction networks as the input prior network, including Harmonizome (Rouillard, et al. 2016), InWeb_InBioMap (Li, et al. 2017), PathwayCommons (Rodchenkov, et al. 2020) and Omnipath (Türei, et al. 2016) (please see Table R2 in the response to Comment 3.4 for the statistics about these networks). We evaluated the resulting performances on both GRN construction (Figure R16) and driver regulator identification (Figure R17) using these different prior gene interaction networks as input. The results demonstrated that different types of gene interaction networks indeed affected the effectiveness of GRN construction and driver regulator identification. Specifically, an input network with larger scale and higher average degree can lead to better performances. The best results were obtained through using the NicheNet network, probably because it covered more interactions. To quantitatively measure the completeness of these prior gene interaction networks, we simply calculated the difference between the number of edges in the prior gene interaction networks (i.e., $|E_{\text{prior_net}}|$) and the number of edges in the ground-truth network (i.e., $|E_{\text{ground_truth}}|$) (Figure R16(c)). The negative values in Figure R16(c) indicated that, even if considering the entire prior gene interaction network, there were still many interactions in the corresponding ground-truth network that cannot be covered, thus tending to yield worse accuracies. Overall, the above results suggested that the performance of our model was related to the completeness of the input network. Please also see our response to Comment 3.4 for more discussions about the input prior networks. **In the revised manuscript, we have added these results to Supplementary Figure 18 and Supplementary Figure 19, and also included the relevant texts in the Discussion section (Line 400-407, Page 12).**

Second, in our original manuscript, we have previously analyzed the effect of edge perturbation of the prior gene interaction network on GRN construction. The results, as shown in Figure R18 (also Supplementary Figure 3 in the original manuscript), suggested that random shuffling of the prior gene interaction network decreased the performance of GRN construction. These results further illustrated the importance of the completeness of the input network.

Finally, to answer the reviewer's question about the contribution of expression versus prior gene interaction networks, we can refer to the comparison results among the non-prior based methods (i.e., GRNBoost2 and DeepSEM), Random_NicheNet and our proposed CEFCON method in Figure R19 in our response to Comment 3.2. More specifically, the non-prior based methods, which construct GRNs only based on gene expression data, had worse performance than prior-based methods in general, suggesting that incorporating prior information does improve the structure recovery. Furthermore, CEFCON had a clear advantage over just randomly selecting edges from the NicheNet network (i.e.,

Random_NicheNet), indicating that gene expression also provides important cell-lineage-specific information. In summary, gene expression provides critical information for understanding the biological significance under a specific scenario, and the prior gene interaction network also greatly benefits GRN construction. The combination of both factors can thus further advance the GRN construction.

Figure R16. Performance evaluation on GRN construction using different prior gene interaction networks on the hESC and mESC datasets. **a**, The performance measured in terms of AUPRC. **b**, The performance measured in terms of EPR. The NicheNet network with only directed edges, i.e., NicheNet (directed only), and the other four baselines, including Harmonizome, InWeb_InBioMap, PathwayCommons, and Omnipath (interactions), were considered for comparison. **c**, The difference between the number of edges in the prior gene interaction networks (i.e., $|E_{\text{prior_net}}|$) and the number of edges in the ground-truth network (i.e., $|E_{\text{ground_truth}}|$).

Figure R17. Performance evaluation on driver regulator identification using different prior gene interaction networks. a, The performance evaluation on the hESC dataset. **b,** The performance evaluation on the mESC datasets. The NicheNet network with only directed edges, i.e., NicheNet (directed only), and the other four baselines, including Harmonizome, InWeb_InBioMap, PathwayCommons, and Omnipath (interactions), were considered for comparison. The precision scores of the top-10 and top-20 predicted genes among all known genes in the four ground-truth gene sets were calculated, respectively.

Figure R18. The effect of edge perturbation of the prior gene interaction network on GRN construction on the mESC and hESC datasets. a, The edge perturbation effect measured in terms of AUPRC. **b,** The edge perturbation effect measured in terms of EPR. The accuracies on the original prior gene interaction network and the ones with five different shuffling ratios were reported. The box-plots results over 20 repeats were reported.

References:

- Rouillard, Andrew D., et al. "The harmonizome: a collection of processed datasets gathered to serve and mine knowledge about genes and proteins." *Database* 2016 (2016).
- Li, Taibo, et al. "A scored human protein–protein interaction network to catalyze genomic interpretation." *Nature methods* 14.1 (2017): 61-64.
- Rodchenkov, Igor, et al. "Pathway Commons 2019 Update: integration, analysis and exploration of pathway data." *Nucleic acids research* 48.D1 (2020): D489-D497.
- Türei, Dénes, Tamás Korcsmáros, and Julio Saez-Rodriguez. "OmniPath: guidelines and gateway for literature-curated signaling pathway resources." *Nature methods* 13.12 (2016): 966-967.

3.2 - The idea of selecting out edges has been used using linear regression settings from single cell RNA-seq (e.g. in CellOracle and Dictys) as well as for bulk data. The authors need to compare to these methods to fully demonstrate the value of their approach. The methods compared either do not use an input network or do it in a way that is much more flexible (e.g. NetRex, which uses the network as a prior), but has the issue of adding potentially more noisy edges.

Response:

We thank the reviewer for the helpful suggestion on expanding our benchmarking tests. According to the reviewer's suggestion, we have added CellOracle in the performance comparison on GRN construction. The results showed that CEFCON performed better than CellOracle on the benchmarking datasets (Figure R19). **In the revised manuscript, we have included the results of CellOracle to Figure 3 (also see Figure R19 in this response) and updated the corresponding analyses in the Results section.**

We would also like to explain that for the other method (i.e., Dictys) mentioned by the reviewer, we were unable to include it in the comparison. Dictys is an interesting method

for constructing cell-type specific and dynamic GRNs based on single-cell multi-omics (scRNA-seq and scATAC-seq) data. However, the data input to Dictys must contain a file of an RNA read count matrix and a bam file of ATAC-seq reads, which limited its application on the existing benchmarking datasets used in our manuscript. In addition, the data processing and the network inference of Dictys are time-consuming (the runtime was also stated in the example notebook from their Github, which can be accessed at <https://github.com/pinellolab/dictys/blob/master/doc/tutorials/short-multiome/>). For these reasons, we did not compare it with our method in this study. Nevertheless, we have discussed and cited this work in the revised manuscript. Although CellOracle also leverages chromatin accessibility data, it has provided base GRNs preprocessed from the chromatin accessibility data of multiple species (which is analogous to the prior gene interaction network we used in this study). Therefore, it was feasible to use only scRNA-seq data as the input for CellOracle, which was consistent with the input requirements of all the compared methods.

Figure R19. Performance evaluation on cell-lineage-specific GRN construction. The performance of GRN construction on different benchmark scRNA-seq datasets, measured in terms of AUPRC (a) and EPR (b), respectively. Six baselines were considered for comparison, including SCINET, GRNBoost2, DeepSEM, NetREX, CellOracle and Random_NicheNet (i.e., randomly selecting edges from the prior gene interaction network). The mean \pm standard deviation scores over 20 repeats were reported.

3.3 - *It seems the authors used NicheNet's network as the input network which was constructed primarily for inferring cell-cell communication networks. Since the authors are interested to infer gene regulatory networks, it might be better to focus on input networks that are directed, e.g. signaling and regulatory, and not include protein-protein interactions.*

Response:

We appreciate the reviewer's suggestion about focusing on the directed regulatory and signaling interactions as the input network. As mentioned by the reviewer, the NicheNet network was originally applied to model cell-cell communication. In particular, NicheNet provides a broad coverage of potential gene relationships containing ligand-receptor interactions, signaling pathways and transcriptional regulatory relationships from more

than 50 data sources. Since we were mainly interested in inferring gene regulatory relationships and studying the intracellular factors of cell fate decisions, we removed the ligand-receptor interactions. However, we still included the undirected associations such as the protein-protein interactions to incorporate more relevant information.

To investigate the impact of considering only directed relations, we have additionally compared the performance of CEFCON on GRN reconstruction and driver regulator identification using the full NicheNet network versus that using only directed edges (please also see Figure R16 and Figure R17 in our response to Comment 3.1). As shown in Figure R16 and Figure R17, considering the undirected edges in the input network achieved better or equivalent performance on most cases of constructing GRNs and identifying driver regulators across the benchmarking datasets.

Here, we further discussed the benefits of considering undirected edges. In fact, in addition to directed TF-target relationships, many undirected interactions, including those involving TF activity modulating enzymes (such as phosphatases and kinases), TF-TF interactions and co-factor binding in transcriptional regulation (Badia-i-Mompel et al., 2023; Göös et al., 2022), are also functionally important. Many of these non-TF-mediated regulation patterns can be expressed as protein-protein interactions, which play important roles in regulating the activity or localization of TFs. Therefore, we believe that incorporating the undirected interactions can provide useful knowledge for elucidating gene regulatory networks and cell fate decisions.

In the revised manuscript, we have added these comparison results to Supplementary Figure 18 and Supplementary Figure 19, and also included the relevant discussion in the Results section (Line 400-407, Page 12). Moreover, we have updated the code of our CEFCON package so that when loading a prior gene interaction network, the users can set a parameter about whether to consider only directed edges or not.

References:

Badia-i-Mompel, Pau, et al. "Gene regulatory network inference in the era of single-cell multi-omics." *Nature Reviews Genetics* (2023): 1-16.

Göös, Helka, et al. "Human transcription factor protein interaction networks." *Nature communications* 13.1 (2022): 766.

3.4 - It is also unclear if the authors are utilizing the ligand-target networks or the prior network that was inferred by collecting interactions from KEGG, Ramilowski et al., Omnipath, PathwayCommons, InWeb_InBioMap, and other models. If the full prior network was used, it would be valuable to show the prior network statistics including the average degree, degree distribution, number of nodes, etc. Similarly it would be helpful to show these properties of the inferred networks.

Response:

We thank the reviewer for the valuable suggestions on supplementing the statistics

information of the prior network and the inferred networks. For the prior network, CEFCON utilizes the comprehensive gene interactions from NicheNet compiled from over 50 sources including KEGG, PathwayCommons, inWeb_InBioMap and Omnipath. The original NicheNet work was used to model cell-cell communication. However, we only considered intracellular factors that affect the cell fates, and therefore the ligand-receptor interactions were removed from the original NicheNet gene interaction network. We have provided the detailed description of the prior gene interaction network in our revised manuscript (please see Section 4.5 in the Methods section of our manuscript).

To fully understand the properties of the employed prior gene interaction network, we reported its basic statistics including the number of source genes, the number of target genes, the number of edges, average degree, density, average clustering coefficient and the slope of degree distribution (Supplementary Table R2). We also provided the above information of other integrated gene networks that have been added for comparative analyses (these comparisons can also be found in our response to Comment 3.1). As shown in Supplementary Table R2, the NicheNet network is more comprehensive and contains much more genes as well as more gene interactions than the other compared gene networks.

We have also reported the properties of the inferred GRNs on the seven benchmarking datasets (Figure R20). In particular, the degree distributions of the inferred GRNs were plotted, and the major network topological properties, including the number of genes, the number of edges, the slope and the R^2 of the degree distribution, and the average clustering coefficient, were listed in the upper right corner of each subfigure in Figure R20. Here, R^2 is the coefficient of determination for the linear regression model to measure how close the data points are with respect to the fitted linear line. The inset in the bottom left corner of each subfigure in Figure R20 provides the degree distribution (also with the slope, R^2 and average clustering coefficient) of the randomized GRN derived from the prior gene interaction network. As shown in Figure R20, the inferred GRNs exhibited scale-free property, i.e., their degree distributions follow a power law, which is the characteristic of most biological networks (Liu et al., 2020). More importantly, the average clustering coefficients of the inferred networks were significantly larger than those of sub-networks randomly derived from the prior gene interaction networks (Figure R20(h)), which was consistent with the previous findings that the large clustering coefficients are an intrinsic feature of biological networks (Liu et al., 2020).

In the revised manuscript, we have added the statistics on the prior gene interaction networks to Supplementary Table 1, and also included the degree distribution and properties of the inferred GRNs to Figure 3(c-d) and Supplementary Figure 4. We have also included the relevant analyses in the Results section (Line 182-187, Page 6).

Figure R20. Degree distributions and major topological properties of the GRNs constructed by CEFCON on the datasets of hESC (a), hHep (b), mESC (c), mDC (d), mHSC-E (e), mHSC-GM (f) and mHSC-L (g), respectively. The x-axis represents the network degree, denoted as k , and y-axis represents the frequency of the network degree k , denoted as $P(k)$. Both k and $P(k)$ are log-transformed. R^2 is the coefficient of determination for the linear regression model to measure how close the data points are with respect to the fitted linear line. The major network topological properties, including the number of genes, the number of edges,

the slope and the R^2 of the degree distribution, and the average clustering coefficient, are listed in the upper right corner of each subfigure. The inset in the bottom left corner of each subfigure provides the degree distributions (also with the slope, R^2 and average clustering coefficient) of the randomized GRN derived from the corresponding prior gene interaction network. **h**, The comparison between the CEFCON constructed GRN and the randomized GRN derived from the prior gene interaction network, in terms of R^2 values and clustering coefficients. Each dot represents one of the seven datasets. Each box plot represents the distribution of the seven datasets, and the whiskers represent the $1.5 \times$ interquartile ranges.

Table R2. Statistics on the prior gene interaction networks tested in this study.

Dataset	Species	#Source genes	#Target genes	#Total genes	#Edges	Avg. degree	Density	Avg. clustering coefficient	Slope of degree distribution (R^2)
NicheNet	Human	18,569	25,332	25,345	5,290,993	417.518	0.008	0.458	-0.953 (0.768)
	Mouse	17,455	18,579	18,579	5,029,532	541.421	0.015	0.420	-0.844 (0.678)
PathwayCommons	Human	16,688	18,689	19,087	1,105,240	115.811	0.003	0.138	-1.324 (0.864)
	Mouse	15,832	17,590	17,834	1,098,689	123.213	0.003	0.137	-1.299 (0.845)
InBioMap	Human	14,462	16,075	17,430	609,015	69.881	0.002	0.147	-1.347(0.877)
	Mouse	13,742	15,350	16,438	554,357	67.448	0.002	0.149	-1.324 (0.864)
Harmonizome	Human	4,319	26,381	26,780	2,985,645	222.976	0.004	0.492	-0.864 (0.607)
	Mouse	3,836	18,002	18,002	2,446,888	271.846	0.008	0.567	-0.751 (0.529)
Omnipath (interaction)	Human	6,182	7,435	8,725	86,248	19.770	0.001	0.197	-1.386 (0.796)
	Mouse	6,154	7,524	8,726	79,693	18.266	0.001	0.197	-1.406 (0.824)

Note: R^2 is the coefficient of determination for the linear regression model to measure how close the data points are with respect to the fitted linear line.

Reference:

Liu, Chuang, et al. "Computational network biology: data, models, and applications." *Physics Reports* 846 (2020): 1-66.

3.5 - The architecture of the CEFCON framework could be better motivated. Although it is explained that the two layers consider the embeddings when a gene is either a source or a target, it might be worth considering what happens if this was encapsulated in a single layer but still using the directionality to define the neighbors.

Response:

We thank the reviewer for this comment. We first would like to point out that we consider genes as both sources and targets in each layer of our two-layered GNN model, which is consistent with the reviewer’s suggestion of encapsulating them in a single layer but using the directionality to define the neighbors.

Specifically, as shown in Figure R21, we use a graph attention neural network with a two-layer architecture, where each layer encompasses two parallel channels to separately capture the embeddings of a gene as either a source or a target. In fact, both the out-going and in-coming networks have the underlying network structure and only differ in the direction of message-passing within the GNN. In the in-coming network, the message-passing direction is consistent with the edges, flowing from the source node to the target

node. Conversely, the direction of message-passing in the out-going network is from the target node to the source node. Then the separate embeddings of source and target nodes are concatenated and encapsulated in a single layer (as commented by the reviewer), and finally passed into the subsequent GNN layer. Please see also the model architecture in Supplementary Figure 1 in our manuscript for more details.

We thank the reviewer for pointing out that those details about our model may not have been clearly described in the original manuscript. To improve the clarity of our model, we have modified Fig. 2(a) of the original manuscript by adding the message-passing directions in the illustration of the employed GNN model. We have also refined and updated the relevant explanation in the Results section (Line 90-93, Page 4) of the revised manuscript.

Figure R21. The gene regulatory network (GRN) construction module. **a**, A two-layer graph neural network (GNN) with multi-head attention mechanism is used for GRN construction in CEFCON. In each layer, the network structure is divided into in-coming and out-going networks based on the directions of message-passing to consider the importance of a gene as a regulator or target. The outputs of both directional networks are then concatenated together. **b**, A contrastive learning model used to train the GNN encoder by maximizing the mutual information (MI) between the gene embeddings and their corresponding summary. The corruption operation produces the negative samples by randomly shuffling the node features. The gene differential expression level (log₂ fold change) is additionally considered as a learnable scalar encoding. After convergence, the attention coefficients are then used to construct the cell-lineage-specific GRN.

3.6 - Figure 3 reports Early Precision which is a fraction of the true positives to the top k predicted edges. It is not clear why the number is greater than 1. The authors also need to explain Supp Fig 2. It is not clear what is accumulated AUPR or EPR which is used for the overall comparison. Please also explain how the error bars were computed (what is different between the 20 repeats).

Response:

We thank the reviewer for this comments. In our original manuscript, Figure 3 reports the Early Precision Ratio (EPR) instead of the Early Precision. EPR is defined as the ratio of true positives among the top- k predicted edges compared to that of a random predictor. The precision of a random predictor is the edge density of the ground-truth network. Therefore, an EPR greater than 1 indicates that the prediction performance is better than the randomized prediction.

Next, we explain more details about “the accumulated AUPRC and the accumulated EPR” in Supplementary Figure 2. Since we evaluated the quality of GRN construction on multiple datasets, the effectiveness of different methods may vary depending on the specific tested dataset. To more intuitively visualize the overall performance of the different methods, we directly summed up the accuracy of each method across all the datasets on the corresponding ground-truths.

Regarding the error bars, given that the baselines and our method generally exhibited certain varied performance due to the random initialization of optimization algorithms such as gradient descent, we conducted evaluations over 20 repeats with different random seeds, and the final results were averaged across these 20 repeats and the error bars indicated the corresponding standard deviations.

As suggested by the reviewer, we have added the above descriptions about the metrics and error bars in the corresponding texts and figure caption in our revised manuscript.

3.7 - The authors reanalyzed the hematopoietic system dataset to obtain cells in the erythroid, granulocyte-monocyte, and lymphoid lineages. However, they did not compare these newly inferred networks to those that were available as part of the benchmark. It would be very beneficial to understand how the reanalysis of the data affected the quality of the inferred edges.

Response:

We thank the reviewer for this comment. In our original manuscript, the data we used for the analyses of hematopoiesis differentiation in Section 2.5 was from the same source as those used in the GRN evaluation process in Section 2.2. In GRN evaluation, all methods were evaluated under the BEELINE framework. Thus, we directly used the BEELINE-processed data and selected the top 1,000 highly variable genes. In Section 2.5, we aimed to analyze the biological significance of the driver regulators related to hematopoietic stem cell development obtained by CEFCON. In particular, we reprocessed the data according to the procedure recommended in SCANPY (Wolf, et al., 2018) and selected the top 3,000 highly variable genes as suggested by Luecken et al. (Luecken, et al., 2019), which can facilitate the subsequent analyses and visualization. Since the performance of GRN construction has been evaluated on the benchmarking datasets in Section 2.2, we did not compare the performance here, but rather focused on the biological significance of the

results obtained by CEFCON and analyzed them in more depth.

Reference:

Wolf, F. Alexander, Philipp Angerer, and Fabian J. Theis. "SCANPY: large-scale single-cell gene expression data analysis." *Genome biology* 19 (2018): 1-5.

Luecken, Malte D., and Fabian J. Theis. "Current best practices in single-cell RNA-seq analysis: a tutorial." *Molecular systems biology* 15.6 (2019): e8746.

3.8 - The notion of precision with regards to the GO terms in figure 5 is misleading. It is simply if regulon has at least one enriched term. Furthermore number of enriched terms may not necessarily translate to more biologically meaningful gene sets.

Response:

We thank the reviewer for the helpful comments. We agree with the reviewer that it is not sufficient to only consider the regulon (i.e., the RGMs in our manuscript) with at least one enriched term. To address this concern, we have supplemented a more rigorous assessment through considering the precision of gene sets with at least 50 enriched GO terms and five enriched KEGG pathways, respectively. In addition, we restricted the evaluation using only the significantly enriched terms that covered at least 20% genes within the gene set. The results indicated that CEFCON still identified a larger proportion of biologically functional regulons compared to SCENIC (Figure R22).

Furthermore, we would like to emphasize that, for a comprehensive evaluation of the identified regulons, we not only considered the gene function enrichment analyses on GO and KEGG, but also investigated whether these identified regulons were able to guide the identification of cell states (please see Figure 5(d) in our original manuscript for more details), as was done in the SCENIC work. Taken together, we believe that these results should provide sufficient evidence to demonstrate the superior capability of CEFCON in identifying biologically meaningful gene sets.

In the revised manuscript, we have added these results to Supplementary Figure 11 and also updated the relevant texts in the Results and Methods sections (Line 290-291, Page 9 and Line 647, Page 22).

Figure R22. Precision of the identified RGMs (regulon-like gene modules) with at least 50 significantly enriched GO_BP terms (a) and at least five significantly enriched KEGG pathways (b) on the five tested datasets, respectively.

3.9 - Its great that the authors compared CEFCON-based prioritization to other methods. However, the comparison again is not quite right because it is not clear whether it is CEFCON's GRN that is helping to get better regulators or is it the specific metrics they used (MFVS and MDS, which is not novel). A more informative comparison would be if they compared MFVS and MDS on SCENIC's network and also on the input network since it had good performance on several benchmarks.

Response:

We thank the reviewer for this constructive suggestion. To better analyze the relative impact of CEFCON's GRN and gene importance metrics on gene identification, we have conducted additional evaluations using the GRNs inferred by SCENIC and the original input network as alternatives to the network constructed by CEFCON (Figure R23). Since the network control-based methods (i.e., MFVS and MDS) cannot obtain the rankings of gene importance, we used the degree centrality to rank candidate genes for both compared approaches after obtaining the candidate driver genes using MFVS and MDS. As shown in Figure R23, CEFCON achieved the best performance in most cases and was stable across all ground-truth gene sets, highlighting the benefits of the CEFCON inferred GRN in identifying influential regulators. We have added these results to Supplementary Figure 6 and also updated the relevant texts (Line 230-232, Page 7-8) in the revised manuscript.

Reference:

Liu, Yang-Yu, and Albert-László Barabási. "Control principles of complex systems." *Reviews of Modern Physics* 88.3 (2016): 035006.

Figure R23. Performance comparison on driver regulator identification using different constructed GRNs with the same network control methods (i.e., MDS and MFVS). a, Performance comparison on the hESC dataset. **b,** Performance comparison on the mESC dataset. The precision scores of the top-10 and top-20 predicted genes among all known genes in the four ground-truth gene sets were calculated, respectively. Regulators identified directly from the priori gene interaction network derived from NicheNet and those identified from the SCENIC-inferred GRNs were used for comparison with the results of CEFCON, respectively. Degree centrality was used for measuring the gene importance of the GRNs constructed by individual compared methods.

Minor:

3.10 - The graph attention framework has been used to infer regulatory interactions, though in an unsupervised setting. The authors need to cite this <https://academic.oup.com/bioinformatics/article/38/19/4522/6663989> and discuss it in the context of their work.

Response:

We thank the reviewer for pointing out this reference. The work mentioned by the reviewer,

i.e., GENELink, utilizes a graph attention framework to infer GRNs and has also been tested on the benchmarking datasets from the BEELINE framework. Our proposed method, CEFCON, also employs a graph attention neural network for GRN inference. Nonetheless, there are some notable differences between the two approaches:

- GENELink consists of two graph attention layers followed by two channels of multilayer perceptrons (MLPs). These two channels of MLPs can encode the feature representations of source and target nodes in the network, respectively. Then the feature representations of gene relationships can be derived from the representations of two related nodes. In addition, GENELink does not consider the different directionalities of the message-passing processes. CEFCON also uses two layers of graph attention neural networks. However, within each layer, it separately considers two opposite directions of message-passing to derive the importance of a gene as a source or target. Moreover, instead of representing the edge features, CEFCON directly obtains the edge weights through the attention coefficients.
- GENELink is a supervised learning approach that heavily relies on the known gene-gene interactions present in the ground-truth networks for model training. In contrast, CEFCON is an unsupervised method that does not require any labeled interactions. In general, the ground-truth datasets are lacking, especially for specific cell types or lineages. Hence, we believe that our unsupervised approach generally would be more useful and can be applied to a wider range of scenarios.

Since GENELink uses the ground-truth gene-gene interactions for supervised learning, for a fair comparison, we did not consider it as a baseline for performance evaluation. However, as a promising graph attention-based method for building GRNs, we have cited this work (Chen, et al., 2022) and discussed it in the revised manuscript (Line 407-410, Page 12).

Reference:

Chen, Guangyi, and Zhi-Ping Liu. "Graph attention network for link prediction of gene regulations from single-cell RNA-sequencing data." *Bioinformatics* 38.19 (2022): 4522-4529.

3.11 - The authors use some adjectives both for their method and other methods, which are not necessary, e.g. "outstanding", or "impressive".

Response:

We thank the reviewer pointing out this issue. In the revised manuscript, we have carefully reviewed and removed these unnecessary adjectives to ensure that our expressions are objective and accurate.

3.12 - The MFVS and MDS metrics are interesting for finding important regulators. It would be helpful to show the comparisons to simpler node importance measures based on centrality.

Response:

We thank the reviewer for the valuable suggestion. We agree with the reviewer that it is crucial to investigate if the current scheme of finding key genes in CEFCON would be more advantageous than the simple node centrality measures. Accordingly, we have additionally compared our method with those using different node centrality measures, including degree centrality, out-degree centrality, in-degree centrality, eigenvector centrality and betweenness centrality, on the CEFCON constructed GRNs of hESC and mESC datasets (Figure R24). As shown in Figure R24, CEFCON achieved the best performance on most of the ground-truth gene sets in both datasets. Among all the compared centrality metrics, out-degree centrality performed the best, which made sense because key TFs generally had higher out-degrees (Gerstein, et al., 2012) (related statistics can be found in Supplementary Table 4 in the revised manuscript).

We further compared the top-20 genes identified by CEFCON with those obtained using the centrality metrics on the same constructed GRNs for the hESC dataset (Figure R25). Our findings revealed that out-degree centrality tended to overlook some important non-TFs, such as *CDH1* and *VIM*. Interestingly, these non-TF genes, which had been previously reported to play crucial roles in embryonic stem cell differentiation, exhibited high rankings through using in-degree centrality, eigenvector centrality and betweenness centrality. On the other hand, our CEFCON framework demonstrated its capability in not only assigning high rankings to the most critical TFs (e.g., *NANOG*, *SOX2* and *POU5F1*), but also ensuring a relatively good rankings for important non-TFs (e.g., *CDH1* and *VIM*), associated with embryonic stem cell differentiation. Overall, we believe that CEFCON can obtain both important TFs and non-TFs more comprehensively. **In the revised manuscript, we have added these results to Supplementary Figure 5 and Supplementary Figure 9, and also updated the relevant texts (Line 228-230, Page 7 and 236-238, Page 8).**

Reference:

Gerstein, Mark B., et al. "Architecture of the human regulatory network derived from ENCODE data." *Nature* 489.7414 (2012): 91-100.

Figure R24. Performance comparison of different node importance metrics on driver regulator identification based on the CEFCON constructed GRNs. a, The performance comparison on the hESC dataset. **b,** The performance comparison on the mESC dataset. Five node centrality metrics, i.e., degree centrality, out-degree centrality, in-degree centrality, eigenvector centrality and betweenness centrality, were used as baselines for comparison. These importance metrics were all calculated based on the same GRN constructed by CEFCON for each dataset. The precision values of the top-10 and top-20 predicted genes among all known genes in the four ground-truth gene sets were calculated, respectively.

Figure R25. The top 20 gene rankings of different node centrality measures, including the influence score of the CEFCON method (a), out-degree centrality (b), in-degree centrality (c), eigenvector centrality (d) and betweenness centrality (e), on the CEFCON constructed GRNs of the hESC datasets.

Reviewer #1 (Remarks to the Author):

The authors have expanded the computational and benchmarking tests, compared the computational efficiency of CEFCON with other methods, and conducted a comparative analysis to highlight the driver regulators uniquely identified by CEFCON, which addressed my concerns. I would recommend publishing the article.

Reviewer #2 (Remarks to the Author):

The authors have answered all my questions and addressed all my concerns. I would not have any further comments.

Reviewer #3 (Remarks to the Author):

The authors have carried out several experiments to address my comments. It is useful to see the dependence of performance on the quality of the prior networks. I am glad they mention how the coverage of the prior networks affects performance in the Discussion.

Response to Reviewers' comments

Reviewer #1 (Remarks to the Author):

The authors have expanded the computational and benchmarking tests, compared the computational efficiency of CEFCON with other methods, and conducted a comparative analysis to highlight the driver regulators uniquely identified by CEFCON, which addressed my concerns. I would recommend publishing the article.

Response:

We would like to sincerely thank the reviewer for the constructive comments which helped us a lot to improve our manuscript.

Reviewer #2 (Remarks to the Author):

The authors have answered all my questions and addressed all my concerns. I would not have any further comments.

Response:

We would like to sincerely thank the reviewer for providing thoughtful comments for us to improve our manuscript.

Reviewer #3 (Remarks to the Author):

The authors have carried out several experiments to address my comments. It is useful to see the dependence of performance on the quality of the prior networks. I am glad they mention how the coverage of the prior networks affects performance in the Discussion.

Response:

We would like to sincerely thank the reviewer for taking the time to review our work. The feedback has been invaluable in improving the clarity and comprehensiveness of our work.